# Reprogramming of translation in yeast cells impaired for ribosome recycling favors short, efficiently translated mRNAs

**Swati Gaikwad[1†], Fardin Ghobakhlou[1†‡], David J Young[1,2], Jyothsna Visweswaraiah[1§], Hongen Zhang[1], Alan G Hinnebusch[1*]**

[1]Division of Molecular and Cellular Biology, Eunice Kennedy Shriver National Institute of Child Health and Human Development, National Institutes of Health, Bethesda, United States; [2]Laboratory of Biochemistry and Genetics, National Institute of Diabetes and Digestive and Kidney Diseases, National Institutes of Health, Bethesda, United States

**\*For correspondence:**
ahinnebusch@nih.gov

[†]These authors contributed equally to this work

**Present address:** [‡]Department of Microbiology, Infectiology & Immunology, Faculty of Medicine, University of Montreal, Montreal, Canada; [§]Pandion Therapeutics, Watertown, United States

**Abstract** In eukaryotes, 43S preinitiation complex (PIC) formation is a rate-determining step of translation. Ribosome recycling following translation termination produces free 40S subunits for re-assembly of 43S PICs. Yeast mutants lacking orthologs of mammalian eIF2D (Tma64), and either MCT-1 (Tma20) or DENR (Tma22), are broadly impaired for 40S recycling; however, it was unknown whether this defect alters the translational efficiencies (TEs) of particular mRNAs. Here, we conducted ribosome profiling of a yeast *tma64Δ/tma20Δ* double mutant and observed a marked reprogramming of translation, wherein the TEs of the most efficiently translated ('strong') mRNAs increase, while those of 'weak' mRNAs generally decline. Remarkably, similar reprogramming was seen on reducing 43S PIC assembly by inducing phosphorylation of eIF2α or by decreasing total 40S subunit levels by depleting Rps26. Our findings suggest that strong mRNAs outcompete weak mRNAs in response to 43S PIC limitation achieved in various ways, in accordance with previous mathematical modeling.

## Introduction

The initiation of translation in eukaryotes commences with assembly of a 43S preinitiation complex (PIC) on the small (40S) ribosomal subunit, containing the ternary complex (TC), comprised of GTP-bound eukaryotic initiation factor two and methionyl initiator tRNA (eIF2·GTP·Met-tRNA$_i$), and additional eIFs (−1, −1A, and −3). The 43S PIC attaches to the 5′ end of mRNA, in a manner stimulated by the eIF4F complex bound to the m$^7$G cap, and scans the mRNA 5′-untranslated region (5′UTR) for an AUG start codon, producing a stable 48S PIC on AUG selection by Met-tRNA$_i$. The GTP bound to eIF2 can be hydrolyzed during scanning, stimulated by GTPase activating protein eIF5, but eIF1 impedes release of inorganic phosphate (P$_i$) at non-AUG codons. Start codon recognition triggers dissociation of eIF1 from the 40S subunit, enabling P$_i$ release from eIF2·GDP·P$_i$ and subsequent dissociation of eIF2·GDP and other eIFs from the 48S PIC. The eIF5B-catalyzed joining of the large (60S) subunit produces an 80S initiation complex (IC) ready to begin the elongation phase of protein synthesis (*Hinnebusch, 2014*; *Hinnebusch, 2017*).

Several different mRNA features influence the rate of translation initiation. The efficiency of 43S PIC attachment to mRNA and subsequent scanning of the 5′UTR are hindered by stable secondary structures, such that long, structure-laden 5′UTRs are generally inhibitory and impose a requirement for RNA helicases associated with eIF4F, including eIF4A and Ded1, to resolve mRNA structures near the cap, within the 5′UTR, and surrounding the start codon (*Sen et al., 2015*; *Sen et al., 2016*; *Gupta et al., 2018*; *Gulay et al., 2020*). The nucleotide sequence immediately surrounding the start

codon—the 'Kozak context'—particularly at the −3 and +4 positions (numbered from the A of AUG (+1)) also influences the efficiency of start codon selection, with A nucleotides upstream of the AUG codons enhancing initiation on yeast mRNAs (*Shabalina et al., 2004*; *Zur and Tuller, 2013*). Binding of eIF4F to the mRNA cap is stimulated by interaction of its eIF4G subunit with the poly(A)-binding protein (PABP) bound to the poly(A) tail, promoting circularization of the mRNA (*Hinnebusch, 2011*). Formation of this 'closed-loop structure' might facilitate reinitiation by ribosomal subunits released from the stop codon of the mRNA following termination of translation and ribosome recycling (*Uchida et al., 2002*). There is evidence that closed-loop formation is favored for shorter mRNAs (*Amrani et al., 2008*), which might help account for the negative correlation between mRNA coding sequence length and translation efficiencies observed in yeast and other eukaryotes (*Arava et al., 2003*; *Thompson et al., 2016*). However, theoretical modeling indicates that shorter mRNAs are capable of more efficient recycling of ribosomal subunits from the stop codon to the initiation codon without a mechanism for promoting circularization via end-to-end interactions (*Fernandes et al., 2017*). The abundance of mRNA is also associated with efficient translation, presumably reflecting the need for mRNAs encoding highly expressed proteins to be both abundant—highly transcribed and stable—and optimized for translation.

Translation initiation can also be modulated by altering the activities of general initiation factors. The function of eIF2 in Met-tRNA$_i$ recruitment is inhibited in response to various stresses by phosphorylation of serine-51 of its α-subunit, as phosphorylated eIF2·GDP acts as a competitive inhibitor of eIF2B, the guanine nucleotide exchange factor (GEF) for eIF2. Inhibition of eIF2B reduces assembly of the TC and thereby decreases formation of 43S PICs, attenuating bulk translation initiation in the cell (*Pavitt, 2018*). Reducing TC levels also stimulates the translation of certain mRNAs by blunting the inhibitory effects of upstream open-reading-frames (uORFs) present in their 5′UTRs (*Hinnebusch, 2005*; *Hinnebusch et al., 2016*). Such uORFs are encountered by scanning 43S PICs first, and their translation generally diminishes the number of PICs that reach the main coding sequences (CDS) downstream owing to recycling of ribosomes following termination at the uORF stop codons. Limited reductions in TC by eIF2α phosphorylation allow uORF start codons to be skipped by scanning PICs with attendant increased translation of the main CDS, by various mechanisms. For yeast *GCN4* mRNA, encoding a transcriptional activator of amino acid biosynthetic genes, translation of the first of four uORFs gives rise to 40S subunits that escape recycling at the uORF1 stop codon and resume scanning downstream. At high TC levels, they quickly rebind TC and reinitiate at uORFs 3 or 4, and are efficiently recycled from the mRNA following uORF translation. When TC levels are reduced by eIF2α phosphorylation, catalyzed by protein kinase Gcn2 on amino acid starvation, a fraction of scanning 40S subunits fails to rebind TC until after bypassing all four uORFs, but then bind TC in time to reinitiate at the main *GCN4* CDS instead, thereby inducing *GCN4* translation (*Hinnebusch, 2005*).

Ribosome recycling entails the splitting and release of 80S ribosomes from mRNA at the stop codon following polypeptide chain termination, and occurs in two stages. Dissociation of the 60S subunit from the 80S post-termination complex (post-TerC) occurs first, catalyzed by ABCE1 (Rli1 in yeast), leaving a 40S subunit with the stop codon in the A site and deacylated tRNA in the P site corresponding to the last translated codon of the ORF (*Young et al., 2015*; *Hellen, 2018*). The second stage of recycling, dissociation of the tRNA and 40S subunit from the mRNA, is stimulated in vitro by eIF2D, or by the heterodimer MCT-1/DENR, which exhibit homology to separate regions of eIF2D (*Skabkin et al., 2010*). eIF2D could also stimulate reinitiation of translation by reconstituted mammalian 40S post-TerCs by releasing the deacylated tRNA and making the P site available for recruiting Met-tRNA$_i$ via the TC (*Skabkin et al., 2013*). Structural analysis shows that human MCT-1/DENR can bind to the 40S decoding center and contact both ends of tRNA$_i$ in a model reinitiation intermediate (*Weisser et al., 2017*), suggesting their likely positions in a 40S post-TerC. Both DENR and MCT-1 have been shown to promote translation reinitiation downstream of uORFs with strong Kozak consensus sequences in *Drosophila* and HeLa cells, although the sensitive uORFs in human cells are extremely short, encoding only one amino acid (*Schleich et al., 2014*; *Schleich et al., 2017*). Other recent findings implicate DENR in promoting reinitiation following uORF translation on certain mRNAs in mouse cells, including that for the CLOCK gene, although the relevant uORFs were not the single-codon variety (*Castelo-Szekely et al., 2019*). Interestingly, in reconstituted mammalian systems, eIF2D and MCT-1/DENR were found capable of substituting for eIF2 in GTP-independent recruitment of Met-tRNA$_i$ to the 40S subunit during primary initiation events on certain

mRNAs where the AUG codon is placed in the P site without prior scanning, including leaderless mRNAs, mRNAs containing an internal ribosome entry site, or mRNAs with unstructured A-rich 5'UTRs (*Dmitriev et al., 2010*; *Skabkin et al., 2010*).

Recently, we showed that yeast mutants lacking the genes for both eIF2D (Tma64) and either MCT-1 (Tma20) or DENR (Tma22) exhibit phenotypes consistent with a broad defect in 40S recycling, including a queuing of translating 80S ribosomes immediately upstream of main CDS stop codons, and increased reinitiation at AUG codons in 3'UTRs. At certain genes, it appeared that unrecycled 40S ribosomes could rejoin with 60S subunits and undergo unconventional 80S reinitiation events in the 3'UTRs (*Young et al., 2018*), a process observed previously for unrecycled 80S post-TerCs in cells depleted of ABCE1/Rli1 (*Young et al., 2015*). The ability of Tma64 or Tma20/Tma22 to promote recycling and block reinitiation was also demonstrated in cell extracts with uORF-containing reporter mRNAs (*Young et al., 2018*). Thus, ostensibly at odds with the findings from mammalian cells where MCT-1/DENR enhance reinititiation following uORF translation, the available evidence suggests that the yeast orthologs of these proteins impede reinitiation by promoting dissociation of 40S post-TerCs from the mRNA. It remains possible, however, that the yeast Tma proteins stimulate, rather than inhibit, reinitiation on particular uORF-containing mRNAs in vivo. It is unknown whether Tma64 or Tma20/Tma22 can substitute for eIF2 in Met-tRNA$_i$ recruitment during primary initiation events on any native mRNAs in yeast cells in the manner described for their mammalian orthologs in vitro (*Dmitriev et al., 2010*; *Skabkin et al., 2010*).

Because ribosome recycling is required to provide 40S subunits needed for assembly of 43S PICs to support new initiation events, we wondered whether defective recycling in yeast cells lacking the Tma proteins would reduce the formation of 43S PICs, which in turn, might differentially affect the translation of mRNAs with different inherent rates of initiation. In this view, changes in translation efficiencies (TEs) observed on depletion of the Tma proteins would result from altered competition among mRNAs for limiting PICs rather than loss of mRNA-specific functions of the Tma proteins in either reinitiation or eIF2-independent primary initiation. To explore this possibility, we conducted ribosome profiling on a *tma64Δ/tma20Δ* double mutant, lacking the yeast orthologs of eIF2D and MCT-1, and compared the TEs of all expressed mRNAs in this mutant to their values in isogenic WT cells. The results indicate that eliminating Tma64/Tma20 leads to a reprogramming of translation that generally favors mRNAs that are more efficiently translated in WT cells and disfavors weakly translated mRNAs. Observing the same general reprogramming of TEs in response to eIF2α phosphorylation in WT cells, or to a reduction in WT 40S subunit levels, we propose that competition among mRNAs for limiting 43S PICs is an important driver of changes in TE in cells impaired for 40S recycling.

## Results

### Identification of mRNAs dependent on Tma64 or the Tma20/Tma22 heterodimer for efficient translation in yeast cells

Previously, we provided evidence that the yeast counterparts of eIF2D or the MCT-1/DENR heterodimer (Tma64 and Tma20/Tma22, respectively) act in ribosome recycling following the termination of polypeptide synthesis at stop codons. Analysis of a yeast *tma64Δ/tma20Δ* double mutant (*tmaΔΔ*) suggested the accumulation of unrecycled 40S post-TerCs stranded at stop codons, with two main consequences: (i) queuing of translating 80S ribosomes immediately upstream of 40S post-TerCs stalled at stop codons; (ii) increased reinitiation by the stalled 40S post-TerCs and attendant translation in adjacent 3'UTRs (*Young et al., 2018*). Recently, profiling of 40S subunits (as opposed to 80S ribosomes) has confirmed the genome-wide accumulation of unrecycled 40S subunits at stop codons in the *tmaΔΔ* mutant (D. Young, S. Meydan-Marks, and N. Guydosh, personal communication). We wondered whether the sequestration of 40S subunits in 40S post-TerCs stranded at stop codons, or otherwise engaged in scanning or translation within 3'UTRs, in the *tmaΔΔ* mutant would reduce their availability to assemble new 43S preinitiation complexes and thereby diminish the relative translational efficiencies (TEs) of certain mRNAs in the cell.

To address this possibility, we used ribosome profiling to measure the relative TEs of all expressed mRNAs in the *tmaΔΔ* mutant compared to the isogenic WT, culturing both strains in synthetic complete (SC) medium. Ribosome profiling entails deep-sequencing of 80S ribosome-

protected mRNA fragments (RPFs, or ribosome footprints) in parallel with total RNA. The ratio of sequencing reads of RPFs summed over the coding sequences to the total mRNA reads for the corresponding transcript provides a measure of TE for each mRNA (*Ingolia et al., 2009*). Owing to normalization of the data for total read number in each library, the RPF and mRNA reads and the calculated TEs are determined relative to the average values for each strain and do not indicate absolute levels of mRNA expression or translation. The RPF and RNA read counts between biological replicates for each strain were highly reproducible (Pearson's r ≈ 0.99) (*Figure 1—figure supplement 1A–D*). The normalized RPF counts from all mRNAs aligned at their stop codons for the WT strain showed the expected three nucleotide periodicity and a peak at the stop codon (*Figure 1A*, blue). As we reported previously for cells cultured in undefined rich medium (YEPD) (*Young et al., 2018*), the corresponding metagene profile for the *tmaΔΔ* mutant in defined (SC) medium differs from the WT profile in exhibiting a peak ~30 nt upstream of the stop codon (*Figure 1A*, green versus blue), attributed to queuing of a translating 80S ribosome behind an unrecycled 40S post-TerCs stalled at the stop codon following peptide release and recycling of the large 60S subunit (*Figure 1A*, schematic). The *tmaΔΔ* mutant also exhibits genome-wide elevated 80S occupancies downstream of stop codons (*Figure 1B*, green versus blue), indicating increased translation reinitiation in 3'UTRs by unrecycled post-TerCs (*Figure 1B*, schematic). Thus, key defects associated with impaired 40S recycling following translation termination observed previously in the *tmaΔΔ* mutant cultured in YEPD (*Young et al., 2018*) were reproduced here in the same mutant grown in SC medium.

Furthermore, we found that the *tmaΔΔ* mutant exhibits a reduced ratio of bulk polysomes to monosomes (P/M) compared to that seen in WT cells (*Figure 1C*, (ii) versus (i)), consistent with a diminished rate of bulk translation initiation. This strain also exhibits an accumulation of halfmer polysomes, evident as shoulders on the 80S monosome and disome peaks in total polysome profiles, not found in the profiles from WT cells (*Figure 1C*, (ii) versus (i); *Figure 1—figure supplement 2* (ii) versus (i)). The usual explanation for halfmer polysomes is a delay in joining of 60S subunits to 48S PICs at AUG start codons, such as conferred by reductions in the production or stability of 60S subunits. In fact, a *tma20Δ* single mutant was found to exhibit an abnormally high ratio of free 40S to free 60S subunits in the manner expected from a 60S biogenesis defect (*Fleischer et al., 2006*). An accumulation of unrecycled 40S subunits at stop codons or in 3'UTRs might also contribute to the halfmer formation we observed in the *tmaΔΔ* mutant; however, the association of such 40S complexes with mRNA might be too labile to withstand sedimentation through sucrose gradients without the crosslinking step employed in 40S subunit profiling (*Archer et al., 2016*).

Interrogating the ribosome profiling data for possible changes in TEs of mRNAs associated with elimination of the Tma proteins, we identified a sizable group of 175 mRNAs showing significant reductions in relative TE (ΔTE*tmaΔΔ*_down), and a cohort of 378 mRNAs exhibiting relative TE increases (ΔTE*tmaΔΔ*_up), in the *tmaΔΔ* mutant versus WT cells (at FDR < 0.05, *Figure 1D*). Importantly, these changes in translation were largely independent of changes in mRNA abundance, as very few mRNAs showing significant TE changes also displayed significant mRNA changes (*Figure 1E*), which is consistent with the idea that eIF2D or MCT-1/DENR directly or indirectly influence the translation of a fraction of cellular mRNAs. Interestingly, the mRNAs belonging to the ΔTE*tmaΔΔ*_up group tend to be highly translated in WT cells, showing a significantly higher median TE compared to all mRNAs (*Figure 2A*, solid green versus black). Consistent with this, these mRNAs exhibit multiple features characteristic of well-translated mRNAs, including short CDS and short 5'UTR lengths (*Figure 2B–C*), higher than average Kozak context scores for nucleotides surrounding the AUG start codons (*Figure 2D*), and greater than average transcript abundance and stability (*Figure 2E–F*). In sharp contrast, the group of ΔTE*tmaΔΔ*_down mRNAs tend to be poorly translated in WT cells in comparison to all mRNAs (*Figure 2A*, dotted versus black); and exhibit all the opposite features of the ΔTE*tmaΔΔ*_up group, which are characteristic of weakly translated mRNAs (*Figure 2A–F*). In accordance with these findings, sequence logos of the context nucleotides surrounding the initiation codons reveal that the ΔTE*tmaΔΔ*_up mRNAs show a higher occurrence of A nucleotides immediately upstream of the AUG codons, especially at the −3 position—a characteristic of the most highly translated mRNAs in yeast (*Zur and Tuller, 2013*)—in comparison to the ΔTE*tmaΔΔ*_down group (*Figure 2—figure supplement 1A* versus *Figure 2—figure supplement 1B*).

In agreement with the foregoing results obtained for the two sets of mRNAs showing TE changes that satisfy a specific FDR threshold of significance, the same trends were observed for the global

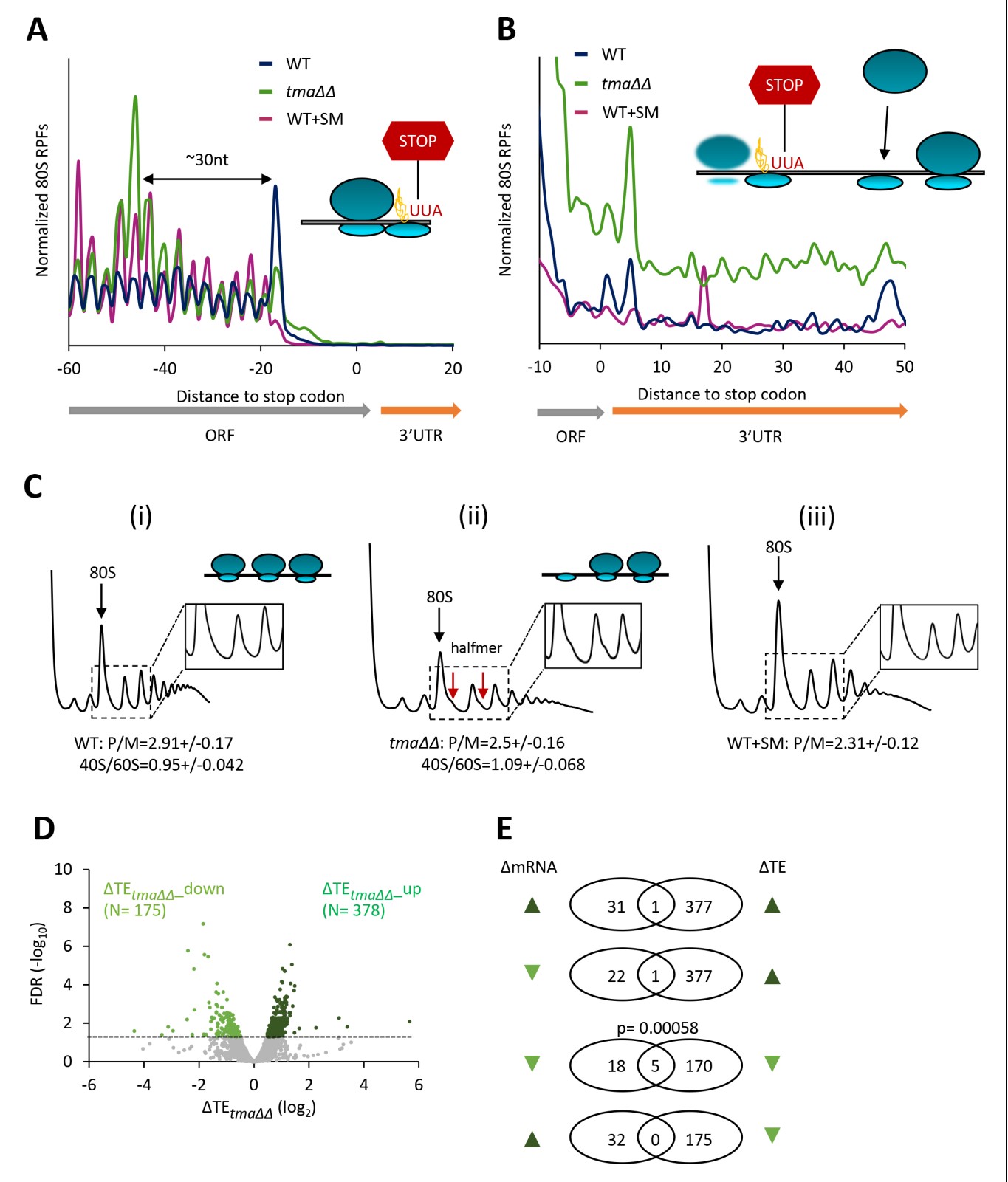

**Figure 1.** Identification of mRNAs exhibiting altered relative TEs on deletion of *TMA64/TMA20*. (**A**) Results from ribosome profiling showing the normalized 80S ribosome reads from all mRNAs aligned with respect to their stop codons, for WT strain BY4741 (blue), the *tmaΔΔ* mutant H4520 (green), and SM-treated WT cells (maroon). Schematics depict queuing of 80S ribosomes 30 nt upstream of the stop codon in the *tmaΔΔ* mutant attributed to 40S post-TerCs stalled at stop codons owing to defective recycling. (**B**) Expanded view of normalized 80S reads from all mRNAs shown in

*Figure 1 continued on next page*

*Figure 1 continued*

(A) for the first 50 nt of the 3'UTRs. The schematic depicts resumed scanning by 40S post-TerCs and reinitiation in 3'UTRs. (C) Polysome profiles of the strains in (A). (i)-(ii) Cells cultured in SC medium at 30°C to log-phase and treated with 50 µg/mL of CHX for 5 min before harvesting at 4°C. (iii) WT cells grown as in (i) except in SC-Ile/Val medium and treated with 1 µg/mL SM for 20 min before CHX addition. WCEs were resolved by sedimentation through sucrose gradients and scanned at 260 nm to visualize (from left to right) free 40S and 60S subunits, 80S monosomes, and polysomes. Tracings are magnified in the insets to show halfmer polysome positions (red arrows) in the *tma∆∆* mutant, depicted in the schematic above (ii). The mean polysome/monosome (P/M) ratios and free 40S to 60S subunits ratios with ± standard error of the means (SEMs) from five biological replicates are shown. (D) Volcano plot showing the $\log_2$ ratios of relative TEs in the *tma∆∆* mutant versus WT cells ($\Delta TE_{tma∆∆}$ values) for each mRNA (x-axis) versus negative $\log_{10}$ of the False Discovery Rate (FDR) for the $\Delta TE_{tma∆∆}$ changes determined by DESeq2 analysis of ribosome profiling data for the 5405 mRNAs with evidence of translation (y-axis). The dotted line marks the 5% FDR threshold. mRNAs showing a significant increase ($\Delta TE_{tma∆∆}$_up) or decrease ($\Delta TE_{tma∆∆}$_down) in relative TE in the *tma∆∆* mutant versus WT cells at FDR < 0.05, are plotted in dark or light green circles, respectively. (E) Venn diagrams of overlaps between differentially expressed mRNAs ($\Delta$mRNA) (FDR < 0.05 and $\log_2\Delta$mRNA >1 or <-1) or differentially translated mRNAs ($\Delta$TE, FDR < 0.05) between the *tma∆∆* mutant versus WT cells, with arrows indicating increased (up) or decreased (down) mRNA or TE in the *tma∆∆* mutant versus WT cells. p values were calculated using Fisher's exact test and were shown only for over-enrichment compared to expectation by chance.

The online version of this article includes the following figure supplement(s) for figure 1:

**Figure supplement 1.** Reproducibility between biological replicates of ribosome footprint profiling and RNA-seq analyses for WT, the *tma∆∆* mutant, and SM-treated WT.

**Figure supplement 2.** Polysome profiles of WT, the *tma∆∆* mutant, and SM-treated WT under the growth conditions used for ribosome profiling and RNA-seq.

TE changes measured for all expressed mRNAs in the *tma∆∆* mutant versus WT cells. Thus, the $\Delta TE_{tma∆∆}$ values for all mRNAs show marked positive correlations with their TE values in WT cells (*Figure 2—figure supplement 2A*, column 6); and significant, albeit weaker, positive correlations with transcript abundance, mRNA half-life and Kozak context scores (*Figure 2—figure supplement 2A*, columns 3–5). A marked negative correlation with $\Delta TE_{tma∆∆}$ values was also observed for CDS length, and a lesser negative correlation was evident for 5'UTR length (*Figure 2—figure supplement 2A*, columns 1–2). Together, these findings suggest that, on deletion of the *TMA64* and *TMA20* genes, mRNAs that are strongly translated in WT cells tend to exhibit an increased relative TE, whereas weakly translated mRNAs tend to show a decrease in relative TE. One way to explain these results is to propose that a competition between 'strong' and 'weak' mRNAs for limiting translation components is skewed in favor of the strong mRNAs in the *tma∆∆* mutant.

## Evidence that relative TE changes observed in the *tma∆∆* mutant do not arise from eliminating an alternative function of eIF2D

As mentioned above, studies in reconstituted mammalian systems revealed that eIF2D can promote binding of Met-tRNA$_i$ to AUG codons in specialized mRNAs in which ribosomal scanning is not required to place the AUG codon in the P site of the 40S subunit. Thus, it was conceivable that the yeast homolog of eIF2D (Tma64) promotes translation initiation on the subset of mRNAs whose TEs are reduced in the *tma∆∆* mutant by complementing the function of eIF2 in delivering Met-tRNA$_i$ to the 40S subunit. The TE reductions for this group of mRNAs might then indirectly stimulate translation of the other group of mRNAs found to be upregulated in the *tma∆∆* mutant owing to reduced competition for initiation machinery. In an effort to rule out this possibility, we conducted ribosome profiling of a *tma64∆* single deletion mutant lacking the yeast homolog of eIF2D but still containing Tma20/Tma22, the homologs of MCT-1/DENR.

The ribosome footprint and RNA-seq results between biological replicates for the WT and *tma64∆* single mutant strains were highly reproducible (Pearson's r ≈ 0.99) (*Figure 3—figure supplement 1A–D*). Polysome profiles of the *tma64∆* single mutant lack the halfmer polysomes observed in the *tma∆∆* mutant (*Figure 3A*, (iii) versus (ii)), as expected if *tma20∆* is responsible for halfmer formation in the double mutant. Also absent in the *tma64∆* single mutant is the queuing of 80S ribosomes upstream of stop codons (*Figure 3B*), and the evidence of genome-wide elevated 3'UTR translation (*Figure 3C*), that are both evident in the *tma∆∆* mutant. Thus, the *tma64∆* single mutant lacks the key phenotypes associated with impaired 40S recycling observed in the *tma∆∆* mutant, suggesting that either the MCT-1/DENR heterodimer is more important than eIF2D for 40S recycling, or that MCT-1/DENR and eIF2D are fully redundant for this activity in yeast. (The

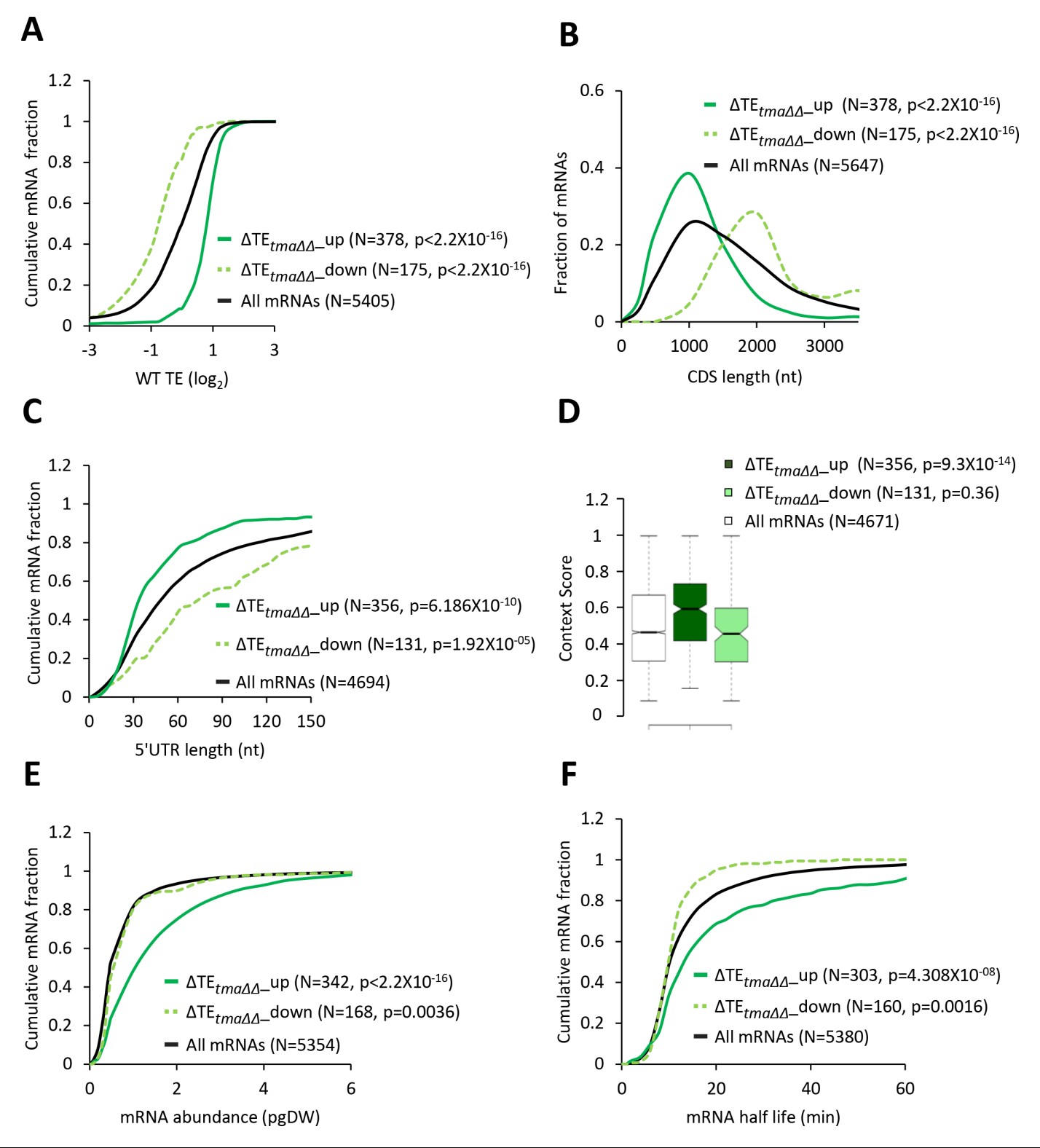

**Figure 2.** mRNAs with increased relative TEs in the *tmaΔΔ* mutant exhibit multiple features of efficiently translated mRNAs. (A, C, E, and F) Cumulative distribution function (CDF) plots of log₂WT TE values (A), 5'UTR length (C), mRNA abundance in molecules per picogram of dry cellular weight (pgDW) (E), and mRNA half-life (F), for all mRNAs (black) and mRNAs exhibiting a significant increase (ΔTE$_{tmaΔΔ}$_up, solid dark green) or decrease (ΔTE$_{tmaΔΔ}$_down, dotted light green) in relative TE (at FDR < 0.05) in the *tmaΔΔ* mutant versus WT cells. (B) Frequency distribution plots of CDS length for the groups of mRNAs examined in (A). p values in panels A-C and E-F were calculated using the Kolmogorov-Smirnov test. (D) Notched box plots of

*Figure 2 continued on next page*

*Figure 2 continued*

context scores calculated for positions −3 to −1 and +4 of main CDS AUGs for all mRNAs (white) and the $\Delta TE_{tma\Delta\Delta}$_up (dark green) and $\Delta TE_{tma\Delta\Delta}$_down (light green) mRNAs examined in (**A**). p values indicated in the panel were calculated using the Mann-Whitney U test. For this and all other box plots below, the upper and lower boxes contain the second and third quartiles and the band gives the median of the data. If the notches in two plots do not overlap, there is roughly 95% confidence that their medians are different.

The online version of this article includes the following figure supplement(s) for figure 2:

**Figure supplement 1.** mRNAs showing relative TE increases or decreases in response to the *tma*ΔΔ mutations, increased eIF2α phosphorylation in WT, or 40S subunit depletion by *rps26*ΔΔ tend to exhibit good or poor Kozak context surrounding their AUG start codons, respectively, compared to all mRNAs.

**Figure supplement 2.** Correlations of genome-wide changes in relative TE evoked by different mutations or conditions with various mRNA attributes.

metagene profile for the WT strain in this experiment (*Figure 3B*) differs from that shown in *Figure 1A* in lacking an 80S peak at the stop codon. In our hands, this feature varies among independent experiments, whereas the queuing of 80S ribosomes ~ 30 nt upstream of stop codons in the *tma*ΔΔ mutant is highly reproducible).

Next, we interrogated the *tma64*Δ single mutant for changes in TE. In contrast to the *tma*ΔΔ mutant, we found no mRNAs exhibiting significant relative TE changes in the *tma64*Δ single mutant versus WT cells (at FDR < 0.05, data not shown). Specifically, the group of 175 mRNAs exhibiting substantially reduced TEs in the *tma*ΔΔ mutant ($\Delta TE_{tma\Delta\Delta}$_down, defined in *Figure 1D*) showed a relatively slight, albeit significant, decrease in median TE in the *tma64*Δ single mutant versus WT (*Figure 3D* column 4), indicating that elimination of yeast eIF2D alone has only a small impact on the translation of these mRNAs. Likewise, the TE increases for the 378 mRNAs in the $\Delta TE_{tma\Delta\Delta}$_up group were relatively small in the *tma64*Δ single mutant versus the *tma*ΔΔ mutant (*Figure 3E*, column 4 versus 3). These findings are at odds with the possibility that elimination of eIF2D, and its possible role as an alternative to eIF2 for recruitment of Met-tRNA$_i$, is responsible for the widespread TE changes observed in the *tma*ΔΔ mutant. However, we cannot eliminate the possibility that eIF2D and the MCT-1/DENR heterodimer are completely interchangeable for eIF2-independent Met-tRNA$_i$ recruitment in vivo such that MCT-1/DENR is sufficient to provide a WT level of this non-canonical pathway in cells devoid of eIF2D. The correlation between recycling defects and marked TE changes observed on comparing the *tma*ΔΔ and *tma64*Δ mutants is consistent with the model that the defect in 40S ribosome recycling conferred by the *tma*ΔΔ mutations contributes to the reprogramming of TEs observed in this double mutant.

## Reducing TC abundance by eIF2α phosphorylation and elimination of 40S recycling factors confers related reprogramming of relative TEs

Given the role of eIF2D and/or MCT-1/DENR in 40S recycling, we hypothesized that sequestering of 40S subunits in post-TerCs and in ribosomes engaged in aberrant 3'UTR scanning or translation in the *tma*ΔΔ mutant would reduce the availability of free 40S subunits for assembling 43S PICs; which would contribute to a skewing of TEs to disproportionately favor strong over weak mRNAs. If this hypothesis is correct, then reducing assembly of 43S PICs by other means might elicit a similar global reprogramming of translation. As a first test of this prediction, we used ribosome profiling to examine the effects on genome-wide TEs rendered by phosphorylation of serine-51 of eIF2α by protein kinase Gcn2. This phosphorylation event is induced by various stresses and is known to reduce assembly of the eIF2·GTP·Met-tRNA$_i$ TC in yeast cells (*Hinnebusch, 2005*). Gcn2 can be activated by limiting biosynthesis of isoleucine and valine by treatment with sulfometuron methyl (SM), an inhibitor of the branched chain amino acid biosynthetic enzyme encoded by *ILV2* (*Jia et al., 2000*).

Judging by the change in polysome:monosome ratio, inducing eIF2α phosphorylation by SM treatment of WT cells confers a reduction in bulk translation initiation similar to that given by the *tma*ΔΔ mutations, but without the appearance of halfmer polysomes seen in untreated *tma*ΔΔ cells (*Figure 1C* and *Figure 1—figure supplement 2*, (iii) versus (ii) and (i)). These same growth conditions were used to conduct ribosome profiling of WT treated with SM and compared to the results described above from isogenic untreated WT cells. The ribosome profiling and RNA-seq results between biological replicates of the SM-treated WT cells were highly reproducible (Pearson's r ≈ 0.99) (*Figure 1—figure supplement 1E–F*). In addition to reduced bulk translation, a second well-known response to the reduction in TC assembly evoked by eIF2α phosphorylation is induction of

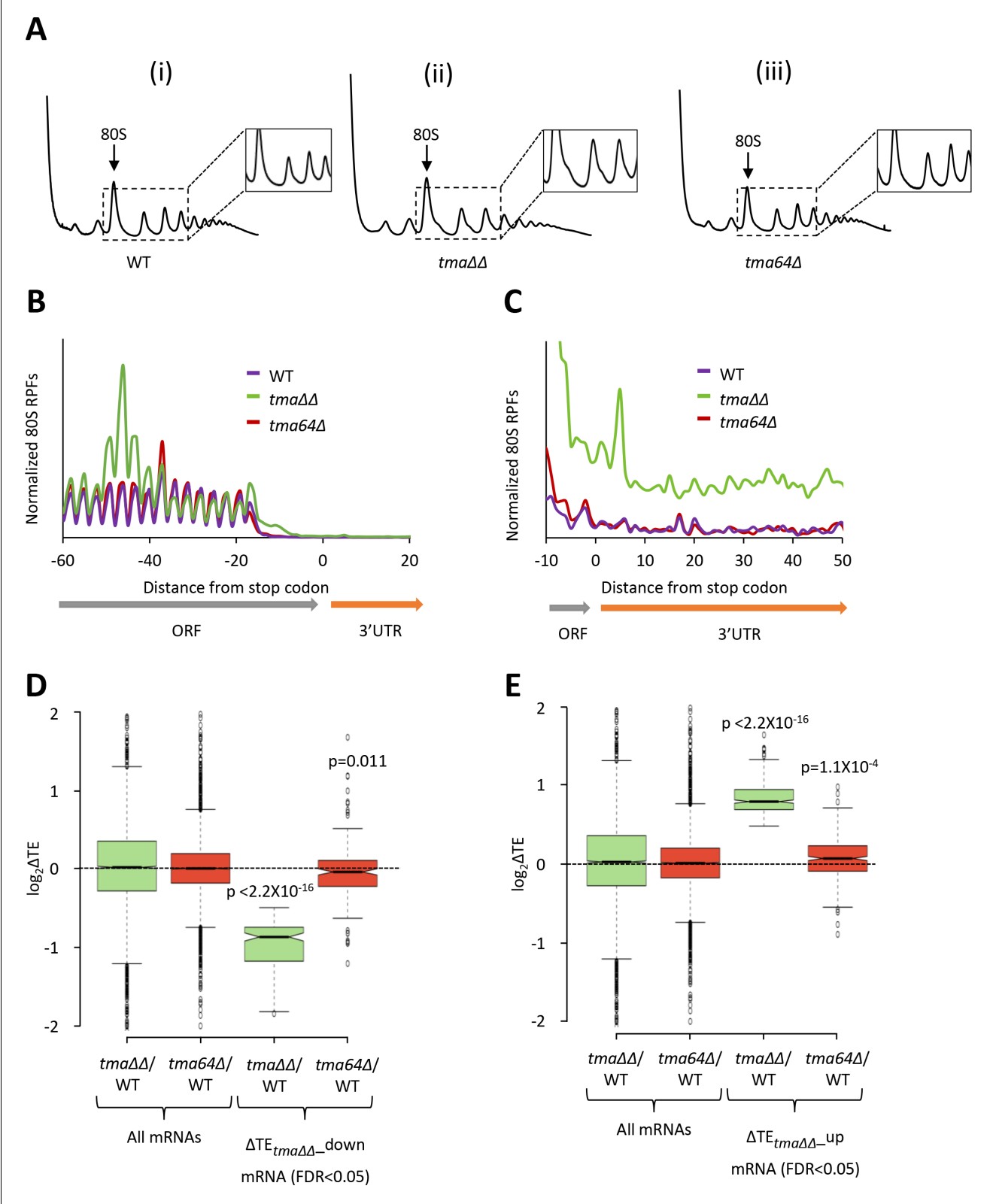

**Figure 3.** Relative TE changes in the *tmaΔΔ* mutant do not result from elimination of eIF2D by the *tma64Δ* mutation. (**A**) Polysome profiles of WT strain BY4741 (i), the *tmaΔΔ* mutant H4520 (ii), and the *tma64Δ* mutant 4051 (iii). Cells were cultured in SC medium at 30°C to log-phase and treated with 50 µg/mL of CHX for 5 min before harvesting at 4°C. WCEs were resolved by sedimentation through sucrose gradients and scanned at 260 nm to visualize (from left to right) free 40S and 60S subunits, 80S monosomes, and polysomes. Tracings are magnified in the insets to show halfmer polysome positions

*Figure 3 continued on next page*

*Figure 3 continued*

in the *tmaΔΔ* mutant (ii). (The polysome profile in (ii) was previously shown in *Figure 1C* (ii)). (B) Results from ribosome profiling showing the normalized 80S ribosome reads from all mRNAs aligned with respect to their stop codons for the WT (purple), the *tmaΔΔ* mutant (green), and the *tma64Δ* mutant (red) described in (A). (Data for the *tmaΔΔ* mutant was plotted previously in *Figure 1A*). (C) Expanded view of normalized 80S reads from all mRNAs shown in (B) for the first 50 nt of the 3'UTRs. (D–E) Notched box plots showing translation changes (log$_2$(ΔTE)) in the indicated mutants versus WT for all mRNAs or for mRNAs exhibiting a significant decrease (ΔTE$_{tmaΔΔ}$_down, N = 175, panel D) or increase (ΔTE$_{tmaΔΔ}$_up, N = 378, panel E) in relative TE in the *tmaΔΔ* mutant versus WT cells at a 5% FDR threshold. A few outliers were omitted from the plots to expand the y-axis scale. p values indicated in panels (D–E) were calculated using the Mann-Whitney U test for the differences between the TE changes for the indicated groups of mRNAs versus all mRNAs.

The online version of this article includes the following figure supplement(s) for figure 3:

**Figure supplement 1.** Reproducibility between biological replicates of ribosome footprint profiling and RNA-seq analyses for WT and the *tma64Δ* mutant.

*GCN4* mRNA translation, owing to the specialized 'delayed reinitiation' mechanism imposed by the four upstream open-reading frames in this transcript (*Hinnebusch, 2005*). Ribosome profiling revealed the expected strong induction of *GCN4* translation evoked by SM treatment of WT cells, as revealed by a large increase in RPF reads in the *GCN4* coding sequence (CDS) with little or no change in *GCN4* mRNA reads, yielding an increase in TE (ΔTE) of ~30-fold (*Figure 4A*, WT+SM versus WT, cf. replicate cultures a and b for Ribo-seq and RNA-seq data). In contrast, no significant increase in *GCN4* translation was observed in the untreated *tmaΔΔ* mutant (*Figure 4A*, the *tmaΔΔ* mutant versus WT), indicating normal levels of TC assembly in this mutant. Furthermore, SM-treated WT cells did not exhibit the queuing of 80S ribosomes upstream of stop codons and elevated 80S occupancies in 3'UTRs observed in the *tmaΔΔ* mutant (*Figure 1A–B*, maroon versus blue), indicating that increased eIF2α phosphorylation does not impair 40S recycling.

Interrogating the ribosome profiling data for changes in TE of specific mRNAs in SM-treated WT versus untreated WT cells (ΔTE$_{WT+SM}$) uncovered 889 and 846 mRNAs whose TEs were relatively upregulated (ΔTE$_{WT+SM}$_up) or relatively down-regulated (ΔTE$_{WT+SM}$_down), respectively, under these conditions (FDR < 0.05, *Figure 4B*). The TE changes evoked by SM treatment were largely independent of changes in mRNA abundance (*Figure 4C*), except that ~ 1/4th of the 889 mRNAs exhibiting increased TEs also show decreased mRNA abundance (*Figure 4C*, 2nd row; more on this phenomenon below). This last finding, plus the considerably larger number of mRNAs whose TEs were significantly changed by SM treatment versus the *tmaΔΔ* mutations, indicate a more widespread reprogramming of TEs evoked by increased eIF2α phosphorylation compared to loss of 40S recycling factors. Nevertheless, as observed for the *tmaΔΔ* mutations, the mRNAs showing increased relative TE on SM treatment (ΔTE$_{WT+SM}$_up) are enriched for mRNAs that are well-translated in untreated WT cells, whereas the ΔTE$_{WT+SM}$_down group of mRNAs are enriched for weakly translated mRNAs (*Figure 4D*). Moreover, the ΔTE$_{WT+SM}$_up group tend to exhibit all of the attributes of strongly translated mRNAs (short CDS and 5'UTR lengths, good Kozak context, greater mRNA abundance and stability, compared to the genome averages); whereas the ΔTE$_{WT+SM}$_down mRNAs are enriched for the features of weakly translated mRNAs (*Figure 4—figure supplement 1A–E*). These opposite features for the ΔTE$_{WT+SM}$_up and ΔTE$_{WT+SM}$_down groups of mRNAs are mirrored by the greater preferences for A nucleotides upstream of the AUG codons of the ΔTE$_{WT+SM}$_up group in comparison to the ΔTE$_{WT+SM}$_down group (*Figure 2—figure supplement 1C* versus *Figure 2—figure supplement 1D*). Furthermore, the global ΔTE values for all expressed mRNAs on SM treatment of WT cells (ΔTE$_{WT+SM}$) are positively correlated with mRNA half-life, transcript abundance, Kozak context score, and the TE in WT cells, while negatively correlated with 5'UTR and CDS lengths (*Figure 2—figure supplement 2B*). As in the case of TE changes in response to the *tmaΔΔ* mutations, the strongest correlations with TE changes in response to SM are for CDS length and for TE in WT cells (*Figure 2—figure supplement 2B* versus *Figure 2—figure supplement 2A*).

Additional support for the idea that eliminating the Tma proteins and increasing phosphorylation of eIF2α produce overlapping alterations in TEs was provided by hierarchical clustering of the TE changes in the two conditions for the group of 486 mRNAs showing significant TE changes, either relative increases or decreases, in response to both the *tmaΔΔ* mutations and SM treatment of WT cells. The results are displayed in the heatmap of *Figure 4E*, where TE increases or decreases for each gene (shown from top to bottom) are indicated by blue or red hues, respectively. It can be

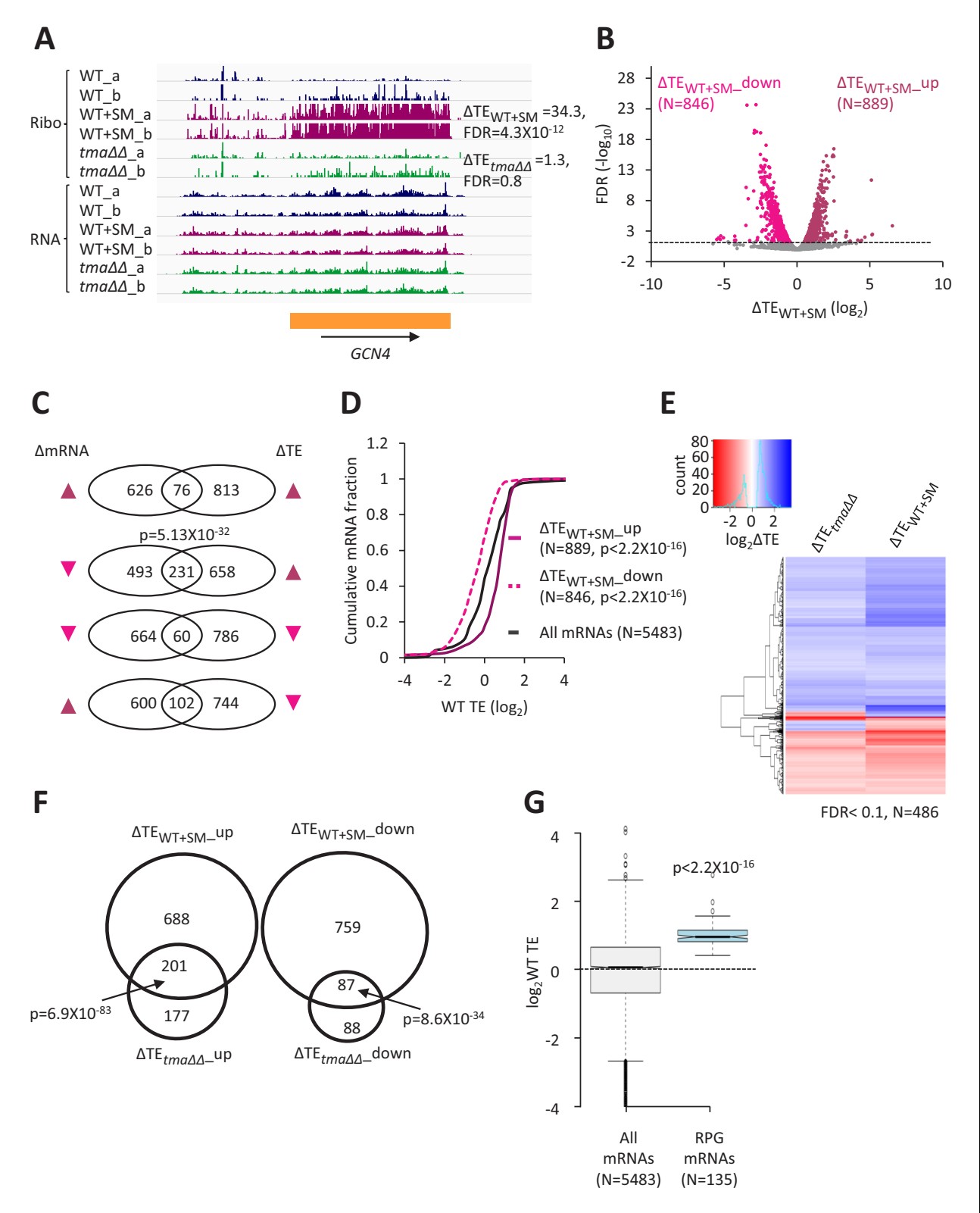

**Figure 4.** Relative TE changes evoked by increased eIF2α phosphorylation overlap substantially with those conferred by deletion of *TMA64/TMA20*. (**A**) Genome browser view of ribosome profiling data for *GCN4* mRNA, showing RPF reads (Ribo) and mRNA reads (RNA) mapping across the transcription unit in WT, with or without SM treatment, and in the *tmaΔΔ* mutant, showing two replicates (**a and b**) for each genotype/condition. The main CDS is shown schematically (orange). The calculated $\Delta TE_{WT+SM}$ and $\Delta TE_{tmaΔΔ}$ values are shown. (**B**) Volcano plot showing log$_2$ ratios of relative TEs in SM-

*Figure 4 continued on next page*

*Figure 4 continued*

treated versus untreated WT cells ($\Delta TE_{WT+SM}$ values) for each mRNA (x-axis) versus negative $\log_{10}$ of the FDR for the $\Delta TE_{WT+SM}$ changes determined by DESeq2 analysis of the 5483 mRNAs with evidence of translation (y-axis). The dotted line marks the 5% FDR threshold. mRNAs showing a significant increase ($\Delta TE_{WT+SM}$_up) or decrease ($\Delta TE_{WT+SM}$_down) in relative TE in SM-treated versus untreated WT cells, at FDR < 0.05, are plotted in dark or light pink circles, respectively. (C) Venn diagrams of overlaps between differentially expressed mRNAs ($\Delta$mRNA) (FDR < 0.05 and $\log_2\Delta$mRNA >1 or <-1) or differentially translated mRNAs ($\Delta$TE, FDR < 0.05) in SM-treated versus untreated WT cells, with arrows indicating increased (up) or decreased (down) mRNA or TE in SM-treated versus untreated WT cells. p values were calculated using the Fisher's exact test and were shown only for over-enrichment compared to expectation by chance. (D) Cumulative distribution function (CDF) plots of $\log_2$WT TE values for all mRNAs (black) and mRNAs exhibiting a significant increase ($\Delta TE_{WT+SM}$_up, solid dark pink) or decrease ($\Delta TE_{WT+SM}$_down, dotted light pink) in relative TE (at FDR < 0.05) in SM-treated versus untreated WT cells. p values were calculated using the Kolmogorov-Smirnov test. (E) Hierarchical clustering analysis of $\log_2$TE changes observed for 486 mRNAs that exhibit significant TE changes in the *tma$\Delta\Delta$* mutant versus WT cells, and also in SM-treated versus untreated WT cells, at FDR < 0.1 for both comparisons. The color scale for $\log_2\Delta$TE values ranges from 2.5 (dark blue) to $-3.5$ (dark red). One mRNA (*YPR146C*) with $\Delta TE_{tma\Delta\Delta}$ >10 fold and $\Delta TE_{WT+SM}$ <10 fold was excluded to enhance the color differences among the remaining mRNAs analyzed in the heatmap. (F) Venn diagrams of overlaps between the groups of mRNAs defined in *Figure 1D* and B showing significantly increased or decreased TEs conferred by SM treatment of WT cells or by the *tma$\Delta\Delta$* mutations. p values were calculated using the Fisher's exact test. (G) Notched box plots showing the $\log_2$WT TEs of the group of 135 RPG mRNAs or all expressed mRNAs. A few outliers were omitted from the plots to expand the y-axis scale. The p value was calculated using the Mann-Whitney U test.

The online version of this article includes the following figure supplement(s) for figure 4:

**Figure supplement 1.** Increased eIF2$\alpha$ phosphorylation in WT upregulates the relative TEs of mRNAs with attributes similar to mRNAs exhibiting relative TE increases in response to the *tma$\Delta\Delta$* mutations.

**Figure supplement 2.** Relative TE changes evoked by increased eIF2$\alpha$ phosphorylation in WT are broadly similar to those conferred by the *tma$\Delta\Delta$* mutations.

**Figure supplement 3.** Gene ontology (GO) analysis of relative TE changes evoked by the *tma$\Delta\Delta$* mutations, increased eIF2$\alpha$ phosphorylation in WT, or the *rps26$\Delta\Delta$* mutations.

**Figure supplement 4.** Ribosomal protein mRNAs are translationally upregulated by the *tma$\Delta\Delta$* mutations, increased eIF2$\alpha$ phosphorylation in WT, or the *rps26$\Delta\Delta$* mutations.

**Figure supplement 5.** Reproducibility between biological replicates of ribosome footprint profiling and RNA-seq analyses for the *gcn4$\Delta$* mutant cultured with or without SM treatment.

**Figure supplement 6.** Condition-specific changes in relative TE evoked by the *tma$\Delta\Delta$* mutations, increased eIF2$\alpha$ phosphorylation in WT, or the *rps26$\Delta\Delta$* mutations.

seen that the majority of mRNAs showing significant TE changes in response to the *tma$\Delta\Delta$* mutations or SM treatment of WT cells exhibit changes in the same direction (same color) in the two conditions (*Figure 4E*); albeit with a tendency for stronger changes (deeper hues) in the SM-treated WT cells. In accordance with these findings, we found highly significant overlaps between the mRNAs in the $\Delta TE_{tma\Delta\Delta}$_up and $\Delta TE_{WT+SM}$_up groups, and between the $\Delta TE_{tma\Delta\Delta}$_down and $\Delta TE_{WT+SM}$_down groups of mRNAs defined above (with changes in TE at FDR < 0.05) on elimination of the Tma proteins or SM treatment of WT cells (*Figure 4F*). Moreover, the median TEs of the groups of mRNAs that are translationally up- or down-regulated by the *tma$\Delta\Delta$* mutations also show significant changes in median TE in the same direction, albeit of diminished magnitude, on SM treatment of WT cells (*Figure 4—figure supplement 2A–B*, columns 3–4). A similar conclusion emerges from comparing the changes in median TE between the two conditions for the groups of mRNAs up- or down-regulated by SM treatment of WT cells (*Figure 4—figure supplement 2C–D*, columns 3–4).

Further evidence that the TEs of many mRNAs change coherently on eliminating the Tma proteins or SM treatment of WT came from gene ontology (GO) analysis of the $\Delta TE_{tma\Delta\Delta}$_up and $\Delta TE_{WT+SM}$_up groups of mRNAs. This exercise revealed that the most significantly enriched functional groups for both sets of upregulated mRNAs are gene products involved in cytoplasmic translation, including ribosome biogenesis (*Figure 4—figure supplement 3A*, cf. maroon and green circles). Indeed, the group of 135 ribosomal protein gene (RPG) mRNAs shows highly significant TE increases in both conditions compared to all mRNAs (*Figure 4—figure supplement 4A and D*). At the same time, the abundance of this group of mRNAs declines under both conditions (*Figure 4—figure supplement 4B and E*), which partially offsets the increases in TEs and dampens the increase in translation levels (i.e. RPF reads uncorrected for mRNA abundance, *Figure 4—figure supplement 4C and F*). The repression of RPG mRNAs in response to starvation for individual amino acids has been described previously (*Natarajan et al., 2001*; *Saint et al., 2014*). The findings above, that the TEs of the RPG mRNAs are upregulated by both the *tma$\Delta\Delta$* mutations and SM treatment of WT cells

(*Figure 4—figure supplement 4A and D*), are in accordance with the fact that they comprise a group of particularly 'strong' mRNAs, exhibiting TEs in WT substantially above the genome average (*Figure 4G*). GO analysis of the $\Delta TE_{tma\Delta\Delta}$_down and $\Delta TE_{WT+SM}$_down groups of mRNAs also revealed common functional categories among those exhibiting the highest enrichments, including protein glycosylation, mannoprotein biosynthesis, and transmembrane transport (*Figure 4—figure supplement 3B*, cf. maroon and green circles).

Given that increased eIF2α phosphorylated induced by SM treatment increases translation of *GCN4* mRNA and alters expression of hundreds of mRNAs owing to enhanced transcriptional activation by Gcn4 (*Jia et al., 2000*; *Natarajan et al., 2001*), the question arose as to whether the transcriptional changes evoked by Gcn4 induction are responsible for the translational reprogramming produced by SM treatment of WT cells. To test this, we conducted ribosome profiling of a *gcn4Δ* mutant treated with SM under the same conditions employed above for WT cells. The ribosome profiling and RNA-seq results between biological replicates for the *gcn4Δ* mutant cultured with or without SM treatment were highly reproducible (Pearson's r ≈ 0.99) (*Figure 4—figure supplement 5A–D*). This experiment probably does not provide a perfect test of whether upregulating Gcn4 contributes to the translational reprogramming evoked by SM treatment because the absence of transcriptional changes produced Gcn4 renders the *gcn4Δ* mutant more sensitive to the growth-retarding effects of Ile/Val starvation by SM. Nevertheless, analyzing the TE changes produced by SM treatment of the *gcn4Δ* mutant revealed a strong reprogramming of TEs with features similar to those described above for SM-treated WT cells and untreated *tmaΔΔ* mutant. In particular, the $\Delta TE$ values for all expressed mRNAs on SM treatment of the *gcn4Δ* mutant show significant positive correlations with TE in WT, transcript abundance, mRNA half-life and Kozak context score, but negative correlations with CDS and 5'UTR lengths (*Figure 2—figure supplement 2D*).

It is striking that eliminating Tma proteins and SM treatment both preferentially increase the TEs of strong mRNAs and disfavor weak mRNAs, and that the majority of mRNAs whose TEs are significantly altered under both of these conditions display changes in the same direction and are enriched for the same functional categories. However, there are also many condition-specific changes in TE that should not be overlooked. Thus, ~50% of the mRNAs whose TEs are upregulated or down-regulated by the *tmaΔΔ* mutations are not significantly altered by SM treatment of WT cells (*Figure 4F*), even though the latter condition generally has a greater impact on the TEs of those mRNAs significantly affected by both conditions (*Figure 4E*). Condition-specific TE changes are also evident in a cluster analysis of TE changes conferred by the two conditions for all 824 mRNAs exhibiting significant TE changes in response to the *tmaΔΔ* mutations (at FDR < 0.1; *Figure 4—figure supplement 6A*). While the TE changes conferred by the *tmaΔΔ* mutations or SM treatment of WT cells are significantly correlated for this group of mRNAs (Spearman's ρ = 0.52, p = 0), there are sizeable blocks of mRNAs that show changes of lesser magnitude (same color, lighter hue), or even changes in the opposite direction (different colors), on SM treatment versus the *tmaΔΔ* mutations (see blocks of mRNAs bracketed in *Figure 4—figure supplement 6A*). Thus, numerous condition-specific changes in TE appear to be superimposed on the shared general trends of increased TEs for strong mRNAs and decreased TEs for weak mRNAs conferred by the *tmaΔΔ* mutations or by SM treatment of WT.

## Relative TE changes evoked by depleting 40S subunits are also related to those conferred by loss of 40S recycling factors and reduced TC assembly

The results thus far suggest that the competition between strong and weak mRNAs is shifted in favor of strong mRNAs when eIF2D and MCT-1/DENR are both eliminated or when eIF2 function is impaired by phosphorylation. Given the role of eIF2D and/or MCT-1/DENR in supplying recycled free 40S subunits for initiation, and the effect of eIF2α phosphorylation in reducing the concentration of TC—a key constituent of initiation complexes—we considered the possibility that both conditions skew the competition among different mRNAs by limiting formation of 43S PICs. If so, then reducing the availability of free 40S subunits by reducing total 40S abundance should likewise diminish 43S PIC assembly and confer a similar reprogramming of TEs. To test this prediction, we depleted the 40S protein Rps26A from a strain lacking the paralog Rps26B by employing a *rps26bΔ* mutant in which *RPS26A* is transcribed from the galactose-dependent, glucose-repressible $P_{GAL1}$ promoter (for brevity, dubbed *rps26ΔΔ*). Polysome profile analysis revealed a marked increase in the ratio of free 60S to 40S subunits 3 hr after shifting the *rps26ΔΔ* mutant to glucose medium, compared to the WT

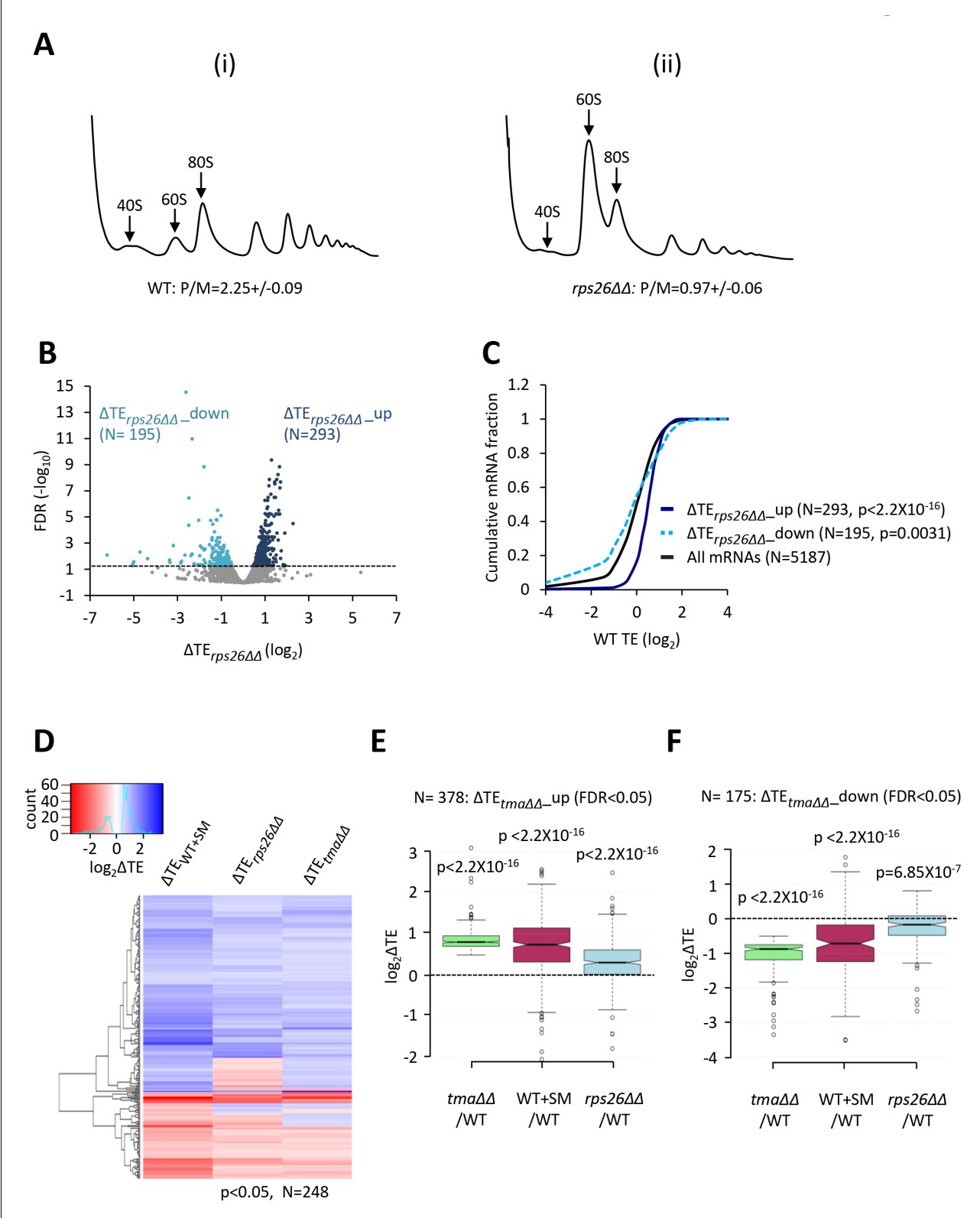

**Figure 5.** Relative TE changes evoked by depleting 40S subunits are broadly comparable to those evoked by the *tmaΔΔ* mutations or by increased eIF2α phosphorylation. (**A**) Polysome profiles of WT strain H2994 and the *rps26ΔΔ* mutant JVY09. Cells cultured in SC medium with 2% galactose and 2% raffinose instead of glucose at 30°C to log-phase were shifted to glucose-containing SC medium for 3 hr and treated with 50 μg/mL of CHX for 5 min before harvesting at 4°C. WCEs were resolved by sedimentation through sucrose gradients and scanned at 260 nm to visualize (from left to right)

*Figure 5 continued on next page*

*Figure 5 continued*

free 40S and 60S subunits, 80S monosomes, and polysomes. The mean polysome/monosome (P/M) ratios ± SEMs from five biological replicates are shown. (**B**) Volcano plot showing the log$_2$ ratios of relative TEs in the *rps26ΔΔ* mutant versus WT cells (ΔTE$_{rps26ΔΔ}$ values) for each mRNA (x-axis) versus negative log$_{10}$ of the FDR for the ΔTE$_{rps26ΔΔ}$ changes determined by DESeq2 analysis of ribosome profiling data for the 5187 mRNAs with evidence of translation (y-axis). The dotted line marks the 5% FDR threshold. mRNAs showing a significant increase (ΔTE$_{rps26ΔΔ}$_up) or decrease (ΔTE$_{rps26ΔΔ}$_down) in relative TE in the *rps26ΔΔ* mutant versus WT cells at FDR < 0.05, are plotted in dark or light blue circles, respectively. (**C**) Cumulative distribution function (CDF) plots of log$_2$WT TE values for all mRNAs (black) and mRNAs exhibiting a significant increase (ΔTE$_{rps26ΔΔ}$_up, solid dark blue) or decrease (ΔTE$_{rps26ΔΔ}$_down, dotted light blue) in relative TE (at FDR < 0.05) in the *rps26ΔΔ* mutant versus WT cells. p values were calculated using the Kolmogorov-Smirnov test. (**D**) Hierarchical clustering analysis of log$_2$TE changes observed for 248 mRNAs that exhibit significant TE changes in the *tmaΔΔ* mutant versus WT cells, in SM-treated versus untreated WT cells, and also in the *rps26ΔΔ* mutant versus WT cells, at p < 0.05 for all genotypes/conditions. The color scale for log$_2$ΔTE values ranges from 3.5 (dark blue) to −2.5 (dark red). (**E–F**) Notched box plots showing translation changes (log$_2$ΔTE) in the indicated mutant versus WT for mRNAs exhibiting a significant increase (ΔTE$_{tmaΔΔ}$_up, N = 385, panel E) or decrease (ΔTE$_{tmaΔΔ}$_down, N = 175, panel F) in relative TE in the *tmaΔΔ* mutant versus WT cells at a 5% FDR threshold. A few outliers were omitted from the plots to expand the y-axis scale. p values were calculated using the Mann-Whitney U test for the differences between the indicated groups of mRNAs versus all mRNAs. (Columns 1–2 in panels (**E–F**) were previously compared in *Figure 4—figure supplement 2A–B*).

The online version of this article includes the following figure supplement(s) for figure 5:

**Figure supplement 1.** Reproducibility between biological replicates of ribosome footprint profiling and RNA-seq analyses for WT and the *rps26ΔΔ* mutant.

**Figure supplement 2.** The *rps26ΔΔ* mutations do not derepress *GCN4* mRNA translation or confer 40S recycling defects.

**Figure supplement 3.** 40S subunit depletion in the *rps26ΔΔ* mutant increases the relative TEs of mRNAs with attributes similar to mRNAs showing TE increases conferred by the *tmaΔΔ* mutations.

**Figure supplement 4.** Effects of the *rps26ΔΔ* mutations on the relative TEs of the "+Rps26" and "ΔRps26" groups of mRNAs.

**Figure supplement 5.** Effects of *rps29bΔ* and *rps0bΔ* mutations on relative TEs of different groups of mRNAs calculated from the ribosome footprint profiling and RNA-seq data of *Cheng et al., 2019*.

strain cultured in parallel (*Figure 5A*), which is diagnostic of a reduction in 40S subunit levels in the mutant. Consistent with this, there was also a reduction in the polysome:monosome ratio in the *rps26ΔΔ* mutant, indicating a reduced rate of translation initiation (*Figure 5A*). Thus, these same growth conditions were used to conduct ribosome profiling of the *rps26ΔΔ* mutant and isogenic WT strain.

The ribosome footprint and RNA-seq results between biological replicates for each strain were highly reproducible (Pearson's r ≈ 0.99) (*Figure 5—figure supplement 1A–D*). Metagene analysis of the ribosome profiling data did not reveal queuing of 80S subunits near stop codons, nor evidence of elevated 3'UTR translation in the *rps26ΔΔ* mutant (*Figure 5—figure supplement 2A–B*). Nor did we observe derepression of *GCN4* translation in the *rps26ΔΔ* mutant versus WT cells; rather, the TE of *GCN4* mRNA was significantly reduced (*Figure 5—figure supplement 2C*), which will be discussed below. Thus, depletion of Rps26 and the attendant reduction in 40S subunit levels should not alter TEs by impairing 40S recycling or reducing TC abundance.

We found that depleting free 40S subunits in the *rps26ΔΔ* mutant upregulates (ΔTE$_{rps26ΔΔ}$_up) and downregulates (ΔTE$_{rps26ΔΔ}$_down) the relative TEs of 293 and 195 mRNAs, respectively, when compared to WT cells (FDR < 0.05; *Figure 5B*). Supporting our prediction, the attributes of the ΔTE$_{rps26ΔΔ}$_up mRNAs were very similar to those translationally stimulated by the *tmaΔΔ* mutations or by eIF2α phosphorylation in SM-treated WT cells: they exhibit greater than average TEs in WT cells (*Figure 5C*, dark blue versus black), Kozak context scores, mRNA half-lives, and mRNA abundance, and shorter than average CDS and 5'UTR lengths (*Figure 5—figure supplement 3A–E*). The translationally repressed group of mRNAs, ΔTE$_{rps26ΔΔ}$_down, showed the opposite trends for at least a subset of these attributes, including lower than average TEs in WT (*Figure 5C*), larger than average CDS lengths (*Figure 5—figure supplement 3A*), and shorter mRNA half-lives (*Figure 5—figure supplement 3D*). In addition, the group of ΔTE$_{rps26ΔΔ}$_up mRNAs show greater enrichment for A nucleotides 5' of the AUG start codons compared to the ΔTE$_{rps26ΔΔ}$_down group, and to all mRNAs (*Figure 2—figure supplement 1E–G*). Examination of the correlations between these mRNA attributes and the global ΔTE changes conferred by the *rps26ΔΔ* mutations for all expressed mRNAs again revealed significant positive correlations with TE in WT, and with transcript abundance, mRNA half-life and Kozak context score, but negative correlations with CDS and 5'UTR lengths (*Figure 2—figure supplement 2C*).

The foregoing results suggest that the TE changes conferred by elimination of Rps26 might overlap substantially with those conferred by elimination of the Tma proteins or by induction of eIF2α phosphorylation by SM treatment of WT cells. Support for this idea is provided by hierarchical clustering of the TE changes for a group of 248 mRNAs showing TE changes in response to all three conditions of the *tmaΔΔ* mutations, SM treatment of WT cells, or depletion of Rps26 in the *rps26ΔΔ* mutant (at $p < 0.05$), where it can be seen that the majority of these mRNAs show TE changes in the same direction for all three conditions (*Figure 5D*). Furthermore, the group of mRNAs whose TEs were upregulated in the *tmaΔΔ* mutant versus WT ($\Delta TE_{tmaΔΔ}$_up), and which also showed an increase in median TE in SM-treated versus untreated WT cells (*Figure 5E*, columns 1–2), likewise displayed an increased median TE in the *rps26ΔΔ* mutant versus WT (*Figure 5E*, column 3). Similar results were observed for the group of mRNAs whose TEs were down-regulated in the *tmaΔΔ* mutant versus WT ($\Delta TE_{tmaΔΔ}$_down), which also showed significant TE reductions in response to both SM treatment of WT cells and the depletion of Rps26 in the *rps26ΔΔ* mutant (*Figure 5F*, columns 1–3). In addition, the mRNAs whose TEs are upregulated by the *rps26ΔΔ* mutations are enriched for the GO categories of ribosomal components and ribosome biogenesis, just as observed for the responses to the *tmaΔΔ* mutations and SM treatment (*Figure 4—figure supplement 3A*, blue versus green/maroon). Consistent with this, the group of 135 RPG mRNAs shows highly significant TE increases in the *rps26ΔΔ* mutant compared to all mRNAs (*Figure 4—figure supplement 4G*). Also, in accordance with the aforementioned results on the *tmaΔΔ* mutant and SM-treated WT, the abundance of the RPG mRNAs declines in response to depletion of Rps26 (*Figure 4—figure supplement 4H*), offsetting the increases in TEs and dampening the increase in translation levels (*Figure 4—figure supplement 4I*).

Despite these commonalities in TE changes, there are sizeable groups of mRNAs whose TEs are affected differently by the *rps26ΔΔ* and *tmaΔΔ* mutations. This is indicated by the relatively smaller changes in median TE conferred by the *rps26ΔΔ* mutations versus the *tmaΔΔ* mutations or SM treatment for the groups of mRNAs whose TEs are up- or down-regulated by the *tmaΔΔ* mutations (*Figure 5E–F*, column 3 versus 1). It is also evident in the cluster analysis of the TE changes conferred by the *rps26ΔΔ* or *tmaΔΔ* mutations for the group of mRNAs showing significant TE changes in response to the *tmaΔΔ* mutations. Although the TE changes for this group of mRNAs are significantly correlated between the *rps26ΔΔ* and *tmaΔΔ* mutations ($ρ = 0.37$, $p = 0$), there are several sets of mRNAs that change by markedly different amounts or in opposite directions in the two mutants (*Figure 4—figure supplement 6B*, see bracketed blocks). Moreover, the $\Delta TE_{rps26ΔΔ}$_down mRNAs do not show significant enrichment for the GO categories shared by the mRNAs whose TEs are down-regulated by the *tmaΔΔ* mutations and SM treatment (*Figure 4—figure supplement 3B*). Thus, as concluded above for the TE changes conferred by SM treatment, there are many TE changes specific to the depletion of Rps26 that appear to be superimposed on the tendency common to all three conditions for strong mRNAs to show increased TEs and weak mRNAs to show reduced TEs; and these differences are generally more pronounced for the *rps26ΔΔ* mutations compared to the *tmaΔΔ* mutations or SM treatment of WT. Below, we consider various explanations for the condition-specific changes in TE, and also explain that the shared trend is in accordance with mathematical modeling of how TEs of different groups of mRNAs should be altered by limiting concentrations of 43S PICs.

### The group of mRNAs with a strong propensity to form the closed-loop intermediate is translationally upregulated by loss of 40S recycling factors, increased eIF2α phosphorylation or 40S subunit depletion

The 5' cap of mRNA is recognized by the cap-binding complex eIF4F, and interaction of the scaffolding subunit eIF4G with the poly(A) binding protein (PABP) enables mRNA circularization. There is evidence that formation of this 'closed-loop' intermediate stabilizes eIF4F binding at the cap (*Park et al., 2011*), and facilitates recycling of 40S subunits from the stop codon to the mRNA 5' end, to increase initiation frequencies (*Uchida et al., 2002*). Using RNA sequencing-immunoprecipitation (RIP-seq) analysis of eIF4F subunits, PABP and the inhibitory eIF4E-binding proteins (4EBPs), Costello et al. identified 'Strong Closed-Loop' (SCL) mRNAs as those enriched in occupancies of eIF4E, eIF4G, and PABP, while de-enriched for association with 4EBPs (*Costello et al., 2015*). This group of SCL mRNAs have shorter than average 5'UTRs and CDS lengths, higher than average Kozak context scores for nucleotides surrounding the AUG start codons, are more abundant, and

more stable compared to all mRNAs, and are highly translated in WT cells (*Figure 6—figure supplement 1A–F*). Noting that these attributes are enriched among the mRNAs whose TEs are elevated by elimination of the Tma proteins, eIF2α phosphorylation, or depletion of Rps26 and 40S subunits, we interrogated the TE changes for the group of SCL mRNAs conferred by each of these three conditions. As shown in *Figures 6A, C and E*, the TEs of the group of SCL mRNAs are significantly elevated in each of the three conditions, whereas the abundance of these mRNAs is significantly reduced under the same conditions (*Figures 6B, D and F*). The inference that limiting 43S PICs upregulates the TEs of SCL mRNAs is also supported by data from Thompson et al., who reported a similar upregulation of SCL mRNAs in strains lacking a ribosomal protein, including *rpp1aΔ* and *rps16bΔ* mutants (*Thompson et al., 2016*). Thus, the SCL mRNAs represent a biochemically defined set of well-translated mRNAs that respond coherently to loss of 40S recycling factors, eIF2α phosphorylation, and a deficit in 40S subunit abundance.

## The *tmaΔΔ* mutations preferentially impair expression of a reporter mRNA harboring a strong cap-proximal secondary structure

We reasoned that if the *tmaΔΔ* mutations reduce the abundance of 43S PICs in the manner suggested above, then they should impair the translation of luciferase reporter mRNAs described previously (*Sen et al., 2016*) harboring insertions expected to form stable secondary structures close to the capped 5′ end predicted to impede 43S PIC attachment. Insertion of a stem-loop (SL) structure of predicted stability of −10.5 kcal/mol five nt from the transcription start site strongly impairs luciferase production in WT cells compared to expression of the parental *LUC* mRNA reporter lacking an insertion, and to the reporter harboring an insertion of lower predicted stability (−5.7 kcal/mol) at the same cap-proximal location (*Figure 6—figure supplement 2*, WT data, rows 1–3). An even greater reduction in luciferase expression in WT cells was conferred by the more stable −10.5 kcal/mol SL inserted at a cap-distal position 44 nt from the 5′ end, and a moderate reduction was observed for the weaker SL inserted at the same location (*Figure 6—figure supplement 2*, WT data, rows 4–5 versus 1), all in agreement with our previous findings (*Sen et al., 2016*). Importantly, the *tmaΔΔ* mutations produced significant reductions in expression for the two reporters harboring cap-proximal SLs, with a particularly strong effect on the reporter bearing the more stable SL (*Figure 6—figure supplement 2*, cf. *tmaΔΔ* versus WT data, rows 2–3 versus 1). In contrast, the *tmaΔΔ* mutations had no significant effects on expression of the two reporters containing the cap-distal SLs, designed to impede ribosomal scanning versus PIC attachment (*Figure 6—figure supplement 2*, cf *tmaΔΔ* versus WT data, rows 4–5), even though both of these SL insertions impaired expression in WT cells. These findings support the idea that eliminating the 40S recycling factors leads to a defect in translation initiation that preferentially affects mRNAs that are inefficiently translated owing to a reduced ability to recruit 43S PICs to the 5′ end.

## Examining the importance of Tma proteins for translational control by uORFs

The foregoing results support a role for Tma proteins in generating free 40S subunits to support 43S PIC assembly through their functions in ribosome recycling. We next examined whether the Tma proteins might be required for reinitiation at downstream CDSs following uORF translation in uORF-containing mRNAs, as noted above for MCT-1 and DENR in animal cells. *GCN4* is the best characterized yeast mRNA exhibiting efficient reinitiation, as ~50% of the 40S subunits that initiate at uORF1 can resume scanning downstream after completing uORF1 translation. Under non-starvation conditions the majority of these 40S subunits reinitiate at uORFs 2–4 and undergo translation termination and ribosome recycling before reaching the *GCN4* AUG. In starved cells where eIF2α is phosphorylated, ~50% of such 're-scanning' 40S subunits can bypass the remaining uORFs 2–4 and reinitiate instead at the *GCN4* AUG (*Hinnebusch, 2005*). If the Tma proteins enhance reinitiation on *GCN4* mRNA, then either single or double deletions of the *TMA* genes should reduce translation of the *GCN4* CDS in starved cells. To address this possibility, we examined a *GCN4-lacZ* reporter containing the native promoter and 5′UTR with all four uORFs intact, which has been used extensively to decipher the delayed reinitiation mechanism of *GCN4* translational control (*Hinnebusch, 2005*). In WT cells, expression of this reporter is low in non-starved cells and increases >10-fold on starvation of WT cells for Ile/Val using SM treatment (*Figure 7A*, columns 1–2). It was previously shown

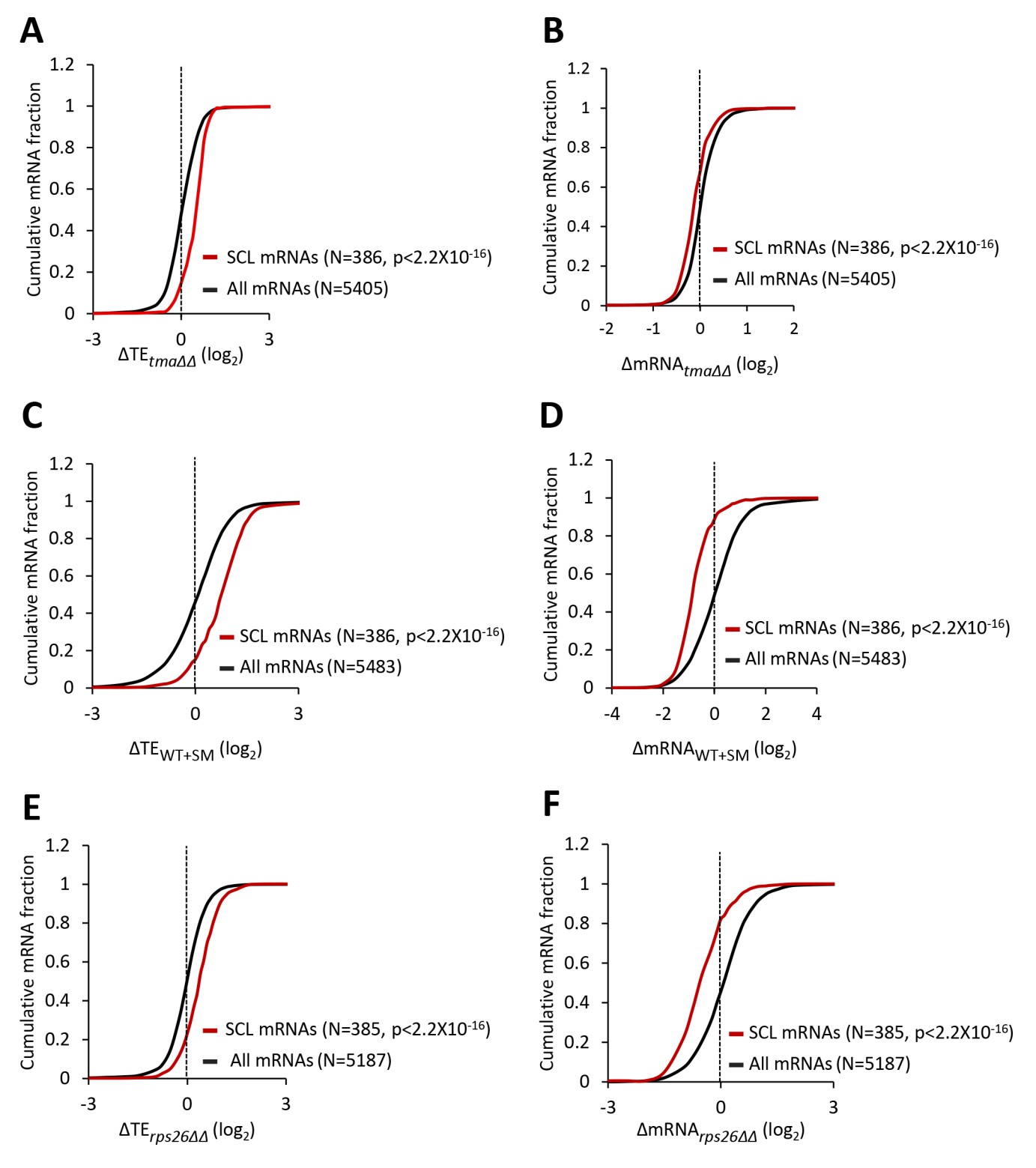

**Figure 6.** mRNAs showing relative TE increases in response to deletion of *TMA64/TMA20,* increased eIF2α phosphorylation, or 40S subunit depletion have a strong propensity to form the closed-loop intermediate. (A–F) Cumulative distribution function (CDF) plots of changes in TEs (log$_2$ΔTE) (A, C, E) or mRNA levels (log$_2$ΔmRNA) (B, D, F) for all mRNAs (black) or the 386 SCL mRNAs (red) in the *tma*ΔΔ mutant versus WT cells (A–B), in SM-treated versus untreated WT cells (C–D), and in the *rps26*ΔΔ mutant versus WT cells (E–F). p values were calculated using the Kolmogorov-Smirnov test. The online version of this article includes the following source data and figure supplement(s) for figure 6:

*Figure 6 continued on next page*

*Figure 6 continued*

**Figure supplement 1.** Strong Closed-Loop (SCL) mRNAs show attributes of highly translated mRNAs.
**Figure supplement 2.** The *tma∆∆* mutations exacerbate the inhibitory effect of cap-proximal stem-loop structures on expression of luciferase reporter mRNAs.
**Figure supplement 2—source data 1.** Source data for results shown in *Figure 6—figure supplement 2*.

that the starvation-induced increased expression of this reporter is completely dependent on the Gcn2 kinase and the uORFs (*Mueller and Hinnebusch, 1986*). Importantly, none of the single or double deletions of the three *TMA* genes produced a significant reduction in *GCN4-lacZ* expression in SM-treated cells, except for a small ~15% reduction in the *tma64∆* single mutant (*Figure 7A*, white bars). The *tma20∆* mutation conferred a ~33% increase in *GCN4-lacZ* expression; however, as the *tma22∆* mutation had no effect, it is difficult to ascribe the modest expression change conferred by the *tma20∆* mutation to inactivation of the Tma20/Tma22 heterodimer. Moreover, the fact that neither the *tma64∆tma20∆* nor *tma64∆tma22∆* double mutant showed significant changes in *GCN4-lacZ* induction (*Figure 7A*, white bars, columns 9–12 versus 1–2) suggests that neither the eIF2D nor MCT-1/DENR yeast orthologs are required for reinitiation at *GCN4* following uORF1 translation when TC levels are reduced by eIF2α phosphorylation.

There was also no effect of the *tma∆∆* mutations on *GCN4-lacZ* expression in non-starvation conditions (*Figure 7A*, black bars), consistent with results in *Figure 4A* for native *GCN4* mRNA. These findings imply that the strong repression exerted by uORFs 3–4 in non-starved WT cells, which should depend on efficient recycling of post-TerCs following termination at these inhibitory uORFs, does not require the Tma proteins. Although this might seem surprising, eliminating the Tma proteins does not confer a recycling defect as severe as that given by depletion of ABCE1/Rli1, which impairs the first step of ribosome recycling (*Young et al., 2015*; *Young et al., 2018*), suggesting that the second step of recycling can occur at appreciable levels without the Tma factors. Presumably, this Tma-independent pathway is sufficient for robust 40S recycling at the stop codons of *GCN4* uORFs 3 and 4.

Going beyond the case of *GCN4*, we next interrogated a previously identified group of mRNAs containing uORFs, initiating with either AUG (262 mRNAs) or one of the nine near-cognate triplets (with a single mismatch to AUG; 1044 mRNAs), which showed evidence of translation in multiple ribosome profiling datasets from various mutant and WT strains (*Zhou et al., 2020*). Based on previous findings that uORFs initiating with AUG codons are more likely to inhibit initiation at downstream CDSs compared to uORFs with near-cognate start codons (*Arribere and Gilbert, 2013*; *Hinnebusch et al., 2016*), we examined the TE changes of the main CDS for the group of 262 mRNAs containing AUG uORFs with evidence of translation in our ribosome profiling data for both the *tma∆∆* and WT strains, and for a second set of 385 mRNAs harboring members of a group of 438 evolutionarily conserved AUG uORFs identified by McManus et al. (*Spealman et al., 2018*). Both groups of mRNAs exhibit significant reductions in CDS TE in the *tma∆∆* mutant versus WT cells (*Figure 7B–C*, columns 4), consistent with the possibility that reinitiation at downstream CDSs following uORF translation is diminished in the *tma∆∆* mutant. However, we found that these two groups of mRNAs also display a lower than average TE in WT cells (*Figure 7D*), and hence, might exhibit TE reductions in the *tma∆∆* mutant owing to reduced competition for limiting 43S PICs. In addition to containing uORFs, these mRNAs could have other features, such as long or structured 5′UTRs, that impede 43S PIC attachment and confer reduced initiation rates when 43S PIC levels decline in the *tma∆∆* mutant. Indeed, both groups of AUG uORF-containing mRNAs also exhibit TE reductions on SM treatment of WT cells or depletion of Rps26 (*Figure 7–C*, columns 5–6), conditions that are not predicted to impair possible functions of Tma proteins in reinitiation downstream of uORFs. Thus, it is difficult to ascribe the TE reductions conferred by the *tma∆∆* mutations for mRNAs with AUG uORFs to diminished reinitiation following uORF translation.

As an orthogonal approach to detecting a role for the Tma proteins in promoting reinitiation, we reasoned that a reduction in reinitiation following uORF translation in the *tma∆∆* mutant should confer an increase in the ratio of RPFs in the uORF versus CDSs, which we termed relative ribosome occupancy (RRO). Applying differential expression analysis using DESeq2 to identify statistically significant changes in RRO values revealed that none of the 843 annotated uORFs initiating with either

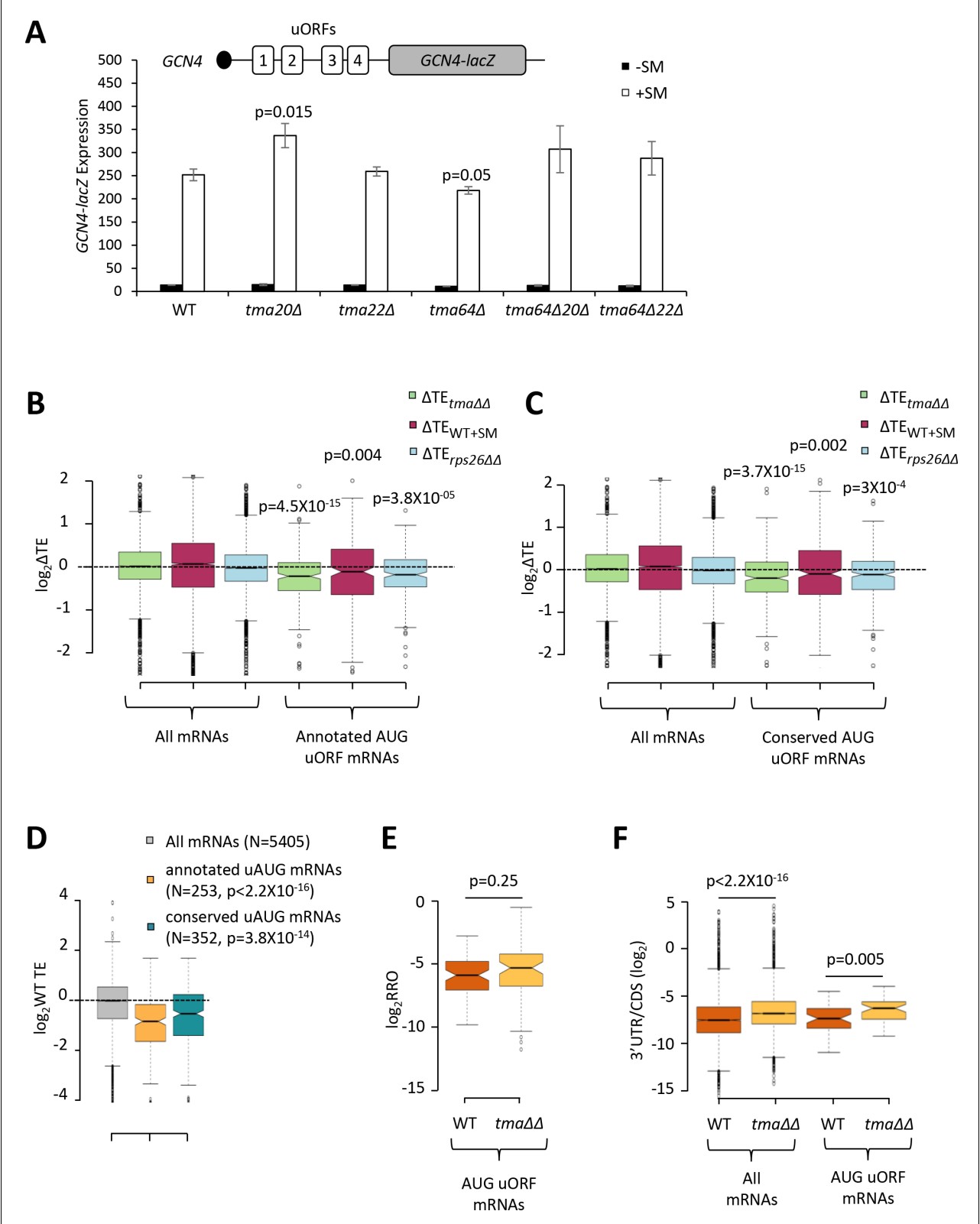

**Figure 7.** Effects of eliminating the 40S recycling factors on translational control by uORFs. (**A**) The *GCN4-lacZ* reporter (depicted schematically) was introduced on plasmid p180 into WT and the indicated yeast mutant strains. Three independently transformed colonies for each strain were cultured in SC-Ura medium at 30°C to log-phase (-SM) or treated with SM at 0.5 μg/mL after 2 hr of growth in SC-Ura and cultured for an additional 4 hr before harvesting. Specific β-galactosidase activities (in units of nmoles of o-nitrophenyl-β-D-galactopyranoside (ONPG) cleaved per min per mg of total

*Figure 7 continued on next page*

*Figure 7 continued*

protein) were measured in WCEs. Error bars are ± SEMs, and p values were calculated using the student's t-test. (**B–C**) Notched box plots showing the log$_2$ΔTE values for all mRNAs or mRNAs containing annotated (**B**) or conserved (**C**) AUG uORFs, conferred by the *tma*ΔΔ mutations (green), SM treatment of WT cells (maroon) or the *rps26*ΔΔ mutations (cyan). A few outliers were omitted from the plots to expand the y-axis scale. p values were calculated using the Mann-Whitney U test for the differences between the AUG uORF-containing mRNAs and all mRNAs for each of the three mutations/conditions. (**D**) Notched box plots showing the log$_2$WT TEs for all expressed mRNAs or for the groups of mRNAs with annotated or conserved AUG uORFs. A few outliers were omitted to expand the y-axis scale. p values were calculated using the Mann-Whitney U test for the differences between the AUG uORF-containing mRNAs and all mRNAs. (**E**) Notched box plot showing log$_2$RRO values for all AUG uORFs and their corresponding main CDSs in WT and the *tma*ΔΔ mutant. A few outliers were omitted to expand the y-axis scale. The p value was calculated using the Mann-Whitney U test. (**F**) Notched box plot showing log$_2$(3'UTR:ORF) ratios for all mRNAs or all AUG uORF-containing mRNAs in WT and the *tma*ΔΔ mutant. To interrogate the same number of mRNAs in WT and the *tma*ΔΔ mutant, one footprint read (arbitrary value) was added to all the reads obtained after aligning the sequence data. A few outliers were omitted to expand the y-axis scale. p values were calculated using the Mann-Whitney U test for the differences between WT and the *tma*ΔΔ mutant.

The online version of this article includes the following source data and figure supplement(s) for figure 7:

**Source data 1.** Source data for results shown in *Figure 7A*.
**Figure supplement 1.** Effects of the *tma*ΔΔ mutations on translational control by uORFs.

AUG or near-cognate start codons, nor any of the evolutionarily conserved translated uORFs identified by McManus et al. (*Spealman et al., 2018*), showing evidence of translation in our WT and *tma*ΔΔ strains exhibited significant changes in RRO in the *tma*ΔΔ mutant versus WT cells, even at a relatively non-stringent FDR of 0.5 (*Figure 7—figure supplement 1A–B*). Furthermore, the group of 63 mRNAs with 70 annotated AUG-initiated uORFs showed no significant difference in median RRO in the *tma*ΔΔ mutant versus WT cells (*Figure 7E*). In contrast, the same group of AUG uORF-containing mRNAs displayed a significant increase in the ratio of RPFs in 3'UTRs versus the main CDS, indicative of increased (not decreased) reinitiation following termination at the main CDS stop codons in the *tma*ΔΔ mutant (*Figure 7F*, column 4 versus 3), which is comparable in magnitude to that seen for all mRNAs (*Figure 7F*, column 2 versus 1). Thus, our results provide no evidence that the Tma proteins enhance reinitiation following uORF translation in yeast cells.

Previous findings on reporter mRNAs in cell extracts suggested that Tma proteins serve to inhibit, rather than promote, reinitiation downstream of uORFs, presumably by stimulating the recycling of 40S post-TerCs at uORF stop codons (*Young et al., 2018*). In this event, a decrease (rather than increase) in RRO might be expected for the AUG uORF-containing mRNAs in the *tma*ΔΔ mutant owing to increased reinitiation at downstream CDS. As shown in *Figure 7E*, a decrease in RRO was not observed in the *tma*ΔΔ mutant. Note, however, that the RPF densities might increase in the uORFs owing to queuing of 80S ribosomes behind the stalled 40S post-TerCs at the uORF stop codons, which could offset an increase in RPFs in the downstream CDSs conferred by enhanced reinitiation, and yield no net change in RRO between the *tma*ΔΔ mutant and WT. If so, this would confound our ability to detect increased reinitiation at the main CDSs conferred by impaired recycling at uORF stop codons by examining RRO changes.

## Discussion

In this study, we set out to determine whether impairing the second step of ribosome recycling by eliminating Tma64/Tma20, and the predicted sequestration of 40S subunits in either stalled 40S post-TerCs, in 43S PICs scanning 3'UTRs, or in 80S ribosomes translating 3'UTR ORFs, has an impact on global TEs in yeast cells. We also wondered whether eliminating Tma64 would alter the TEs of any mRNAs owing to loss of its possible function as an alternative to eIF2 for recruiting Met-tRNA$_i$ to AUG start codons in the manner demonstrated in vitro for its mammalian counterpart, eIF2D. We found that the *tma*ΔΔ mutant lacking both Tma64 and Tma20 exhibits reduced bulk polysome assembly, consistent with a reduced rate of translation initiation on many mRNAs; and by ribosome profiling we observed altered TEs of specific mRNAs, with sizable groups undergoing increases or decreases in relative TE compared to their values in isogenic WT cells. These groups of upregulated and down-regulated mRNAs were enriched for diametrically opposed properties. The mRNAs translationally upregulated in the *tma*ΔΔ mutant tend to be well-translated in WT cells and to have short CDS and 5'UTRs, strong Kozak AUG contexts, to be unusually abundant and to have longer than

average half-lives—all attributes of well-translated 'strong' mRNAs. The mRNAs translationally down-regulated in the *tmaΔΔ* mutant tend to be poorly translated in WT cells and to exhibit all of the opposite properties characteristic of 'weak' mRNAs. The *tma64Δ* single mutant, in contrast, displays none of the hallmarks of impaired 40S recycling, shows no reduction in bulk polysome assembly, and exhibits no statistically significant changes in TE for any mRNAs in our profiling experiments. These results suggest that the translation changes in the *tmaΔΔ* mutant are associated with defective ribosome recycling rather than the absence of Tma64/eIF2D per se and any possible role it might play in Met-tRNA$_i$ recruitment in yeast. However, we cannot rule out the possibility that MCT-1/DENR can substitute completely for eIF2D in providing a WT level of eIF2-independent Met-tRNA$_i$ recruitment in the *tma64Δ* single mutant, and that complete loss of this function in the *tmaΔΔ* mutant contributes to the reprogramming of TEs.

Strikingly, the pattern of TE changes conferred by the *tmaΔΔ* mutations was also evident in the translational reprogramming that accompanies induction of eIF2α phosphorylation by Gcn2 on starvation for amino acids Ile/Val, and also that conferred by depletion of essential 40S subunit protein Rps26. In particular, the group of highly translated 'SCL' mRNAs with a heightened propensity for closed-loop formation, owing to their elevated occupancies of eIF4F and PABP, and the group of RPG mRNAs that are strongly enriched in the SCL group (*Figure 6—figure supplement 1G*), tend to show increased TEs in all three conditions of *tmaΔΔ* mutations, SM treatment of WT cells, and depletion of Rps26. There was also a significant enrichment for 'weak' mRNAs among those down-regulated in all three conditions. These same trends were evident on SM treatment of the *gcn4Δ* mutant, showing that they occur independently of the widespread transcriptional changes conferred by Gcn4 in SM-treated WT cells. Given that eIF2α phosphorylation reduces TC assembly, a key component of 43S PICs, and that both the *tmaΔΔ* mutations and depletion of Rps26 reduce the abundance of free 40S subunits available for PIC assembly, we suggest that the reduced concentration of 43S PICs that should exist in each case is a key factor that skews the competition between strong and weak mRNAs to favor strong mRNAs, and that this altered competition is an important driver of TE changes common to all three conditions. Supporting the idea that the *tmaΔΔ* mutations impair translation initiation by limiting the availability of 43S PICs, they preferentially reduced expression of *LUC* reporter mRNAs harboring stem-loops inserted close to the mRNA 5' end, which are expected to impair 43S PIC attachment, compared to reporters bearing cap-distal insertions or the parental reporter lacking a stem-loop insertion.

We note that the *tmaΔΔ* mutant does not exhibit a reduction in the ratio of free 40S to 60S subunits (*Figure 1C* (ii) versus (i)), ostensibly at odds with our suggestion that the 40S recycling defect in this mutant limits the availability of free 40S subunits for 43S PIC assembly. It is expected that a fraction of the unrecycled 40S subunits will have moved from stop codons and are scanning the 3'UTR to generate the increased 3'UTR translation previously demonstrated in the *tmaΔΔ* mutant (*Young et al., 2018*) and observed here (*Figure 1B*). Such 40S post-TerCs, located either at stop codons or scanning in 3'UTRs, may dissociate from mRNA during polysome fractionation by sucrose density gradient centrifugation because they lack base-pairing between a bound tRNA and an mRNA codon—a property which necessitated the use of formaldehyde fixation to capture scanning 40S subunits in 40S profiling experiments (*Archer et al., 2016*; *Sen et al., 2019*). If so, this dissociation would prevent us from observing the predicted reduction in free 40S subunits. We have also proposed that sequestration of 40S subunits in the 80S ribosomes engaged in abnormal 3'UTR translation in the *tmaΔΔ* mutant, which occurs on essentially all 3'UTRs (*Young et al., 2018*), also contributes to the reduced availability of free 40S subunits for PIC assembly in this mutant; and this sequestration also will not reduce the free 40S/60S ratio. Finally, the *tma20Δ* deletion was shown to confer an elevated free 40S/60S ratio, which may result from a 60S biogenesis defect (*Fleischer et al., 2006*) and this could offset the depletion of free 40S subunits expected from accumulation of unrecycled 40S subunits at stop codons and in 3'UTRs in the *tmaΔΔ* mutant. Nevertheless, we cannot eliminate the possibility that the impaired ribosome recycling in the *tmaΔΔ* mutant is not the major defect responsible for the reprogramming of translation with features characteristic of a reduction in 43S PIC assembly. As noted above, the complete loss of eIF2-independent Met-tRNA$_i$ recruitment could be involved, or a defect in ribosome biogenesis that results in aberrant 40S subunits in addition to reduced 60S subunit levels might exist, either of which could be exacerbated by defective 40S recycling to confer the reprogramming of TEs observed in the *tmaΔΔ* mutant.

The conclusion that conditions expected to reduce the concentration of 43S PICs favor more efficiently translated mRNAs is in full agreement with Lodish's predictions based on mathematical modeling of the kinetics of protein synthesis (*Lodish, 1974*). A key prediction of this model is that reducing the concentration of 43S PICs will alter the TEs of particular mRNAs to different degrees depending on their inherent rates of assembling an 80S initiation complex at the start codon from mRNA and the 43S PIC (assigned an overall rate constant of $K_1$ in the model). On reducing 43S PIC levels, 'weaker' mRNAs (with lower $K_1$ values) will exhibit a proportionally greater reduction in translation compared to 'stronger' mRNAs (with higher $K_1$ values). Considering the relative TE values provided by genome-wide ribosome profiling, the model predicts that weak mRNAs will exhibit a decrease in TE relative to the average mRNA, while strong mRNAs will show an increase in relative TE on reduction of 43S PIC levels, which corresponds to the trends we observed for TE changes in the *tmaΔΔ* mutant, SM treatment of WT, and depletion of Rps26 (*Figure 8*). The Lodish model is supported by several studies of global translation; and other analyses suggest that reductions in 40S availability can also confer increases in absolute translation rates of very 'strong' mRNAs owing to reduced ribosome crowding and collisions during elongation (*Mills and Green, 2017*).

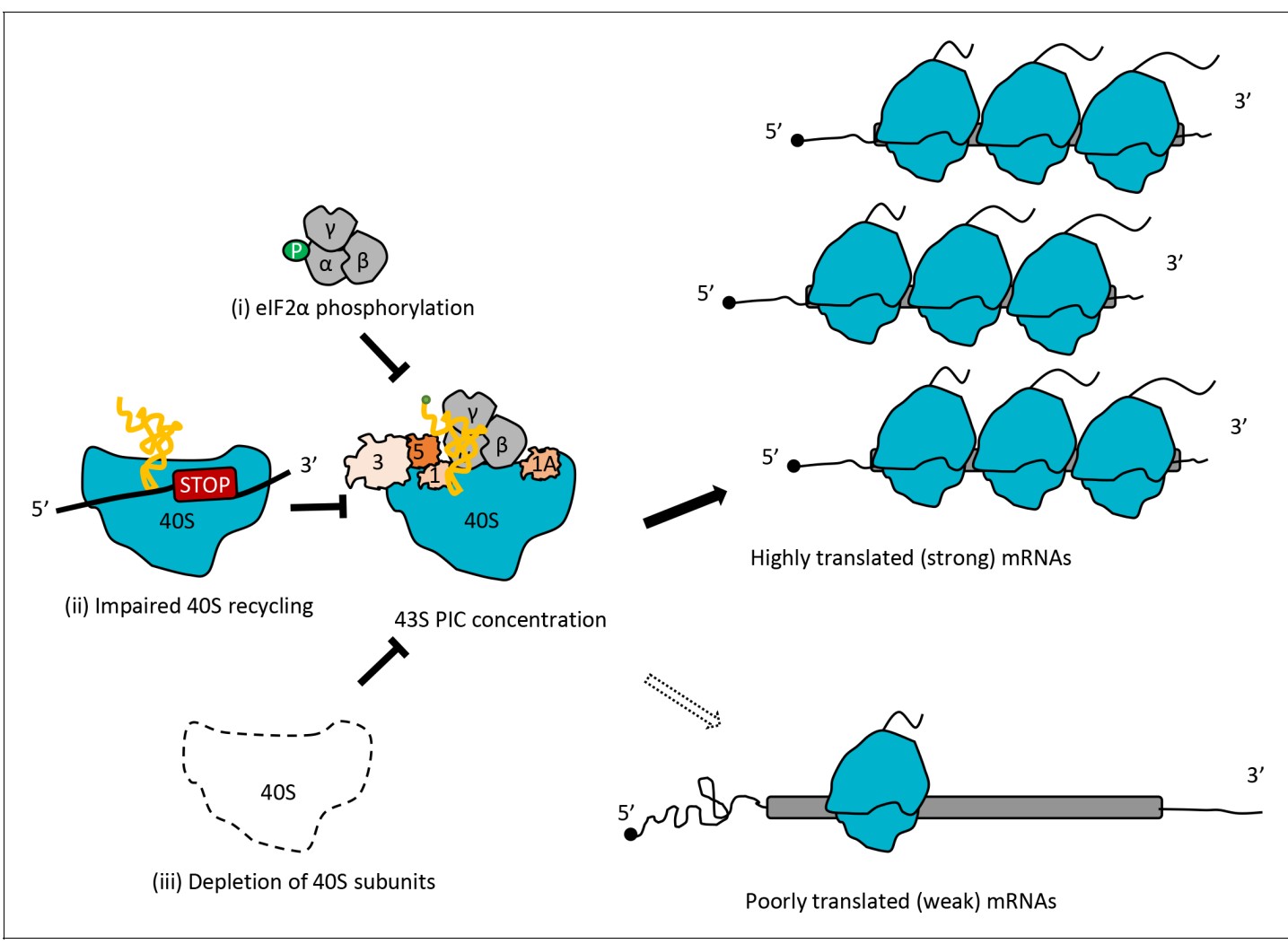

**Figure 8.** Model depicting how impaired 40S recycling, increased eIF2α phosphorylation, or 40S subunit depletion results in reprogramming of TEs by decreasing 43S PIC assembly. Reducing assembly of 43S PICs in any of three different ways, (i) impairing recycling of 40S post-TerCs by the *tmaΔΔ* mutations, (ii) inducing phosphorylation of eIF2α by SM treatment of WT cells, or (iii) decreasing 40S subunit levels by depleting Rps26, skews the competition between inherently 'strong' and 'weak' mRNAs to favor translation of the 'strong' mRNAs, which have higher TEs in WT cells and tend to have shorter, less-structured 5'UTRs and shorter CDSs, compared to 'weak' mRNAs.

It is formally possible that CDS length is a more fundamental driver of TE changes than the basal TE in WT cells under the three mutant/stress conditions analyzed here, but this would beg the question of the molecular mechanism involved. Shorter mRNAs might be more capable of forming the closed-loop intermediate, which could enhance eIF4F association owing to eIF4G-PABP interaction, or more capable of recycling of ribosomes in cis from the stop codon to start codon; but both features would enhance their basal TE and help them compete for limiting 43S PICs at the initiation step. It is also important to note that all the features known to be associated with high basal TE in WT cells are also associated with increased TE in our three mutants/conditions, including elevated propensity for closed-loop formation, higher than average Kozak context scores, higher than average mRNA abundance, and longer than average mRNA half-lives (*Costello et al., 2015*; *Weinberg et al., 2016*). Given the highly conserved inverse relationship between CDS length and TE across many different species (*Arava et al., 2003*; *Thompson et al., 2016*), it is likely that our finding that a short CDS is a predictor of enhanced TE in all three conditions we examined reflects the fact that mRNAs with short CDSs tend to be inherently stronger mRNAs and thus better able to compete for limiting PICs during initiation in the manner predicted by the Lodish model.

While the same trends in TE reprogramming were evident in all three mutant/stress conditions, there were also extensive condition-specific differences among the particular mRNAs being affected, and in the magnitude of changes observed for the same mRNAs. Thus, it seems clear that particular mRNAs respond to different extents, or even in different directions, to eIF2α phosphorylation induced by SM treatment, to the *tmaΔΔ* mutations, or to the reduction in total 40S abundance in the *rps26ΔΔ* mutant. We envision that these condition-specific responses are superimposed on the reprogramming of TEs imposed by increased competition for limiting 43S PICs to sculpt the overall profile of TE changes across the translatome in different ways.

This condition-specificity of TE changes is exemplified by translational regulation of *GCN4*. The TE of *GCN4* mRNA was strongly induced only by SM treatment, which likely reflects the exquisite sensitivity to TC levels of the delayed reinitiation mechanism via uORFs that governs *GCN4* translation. In this mechanism, a slowed rate of TC binding to 40S subunits scanning the *GCN4* mRNA leader following translation of uORF1, and not the availability of free 43S PICs per se, leads to the strong induction of *GCN4* translation when TC concentrations decline on eIF2α phosphorylation (*Hinnebusch, 2005*). It is possible that TE increases observed for certain other mRNAs specifically on SM treatment also result from overcoming the inhibitory effects of uORFs on downstream translation via increased eIF2α phosphorylation. This might include mRNAs with single inhibitory uORFs in addition to the more elaborate set of four regulatory uORFs found in *GCN4,* as reducing TC levels by eIF2α phosphorylation has been shown to mitigate the inhibitory effects of single uORFs (*Young and Wek, 2016*). In the well-studied case of antizyme inhibitor mRNA, in which ribosome stalling occurs during translation of an inhibitory uORF, evidence indicates that eIF2α phosphorylation decreases the rate of 43S PIC loading at the mRNA 5′ end and thereby suppresses queuing of PICs upstream of the uORF start codon to allow a greater proportion of scanning ribosomes to bypass the uORF and translate the downstream CDS (*Ivanov et al., 2018*). In addition to the translation of uORF-containing mRNAs being upregulated by eIF2α phosphorylation via increased reinitiation, it is also possible that primary initiation occurs more efficiently at low TC levels on certain mRNAs (*Dasso et al., 1990*), possibly related to the RNA-binding activity of eIF2β (*Laurino et al., 1999*) or to the role of eIF2 in promoting the recruitment of other eIFs to the PIC in the multifactor complex (*Hinnebusch, 2011*), and that such mRNAs would not be similarly upregulated by the diminished free 40S subunit levels predicted for the *tmaΔΔ* or *rps26ΔΔ* mutations.

One way to account for TE changes conferred specifically by elimination of the Tma factors might be to propose that mRNAs differ in their dependence on the recycling of 40S post-TerCs from the stop codon to the initiation codon of the same mRNA for reinitiation. Indeed, there are theoretical predictions that short mRNAs achieve their relatively higher TEs compared to long mRNAs by this feedback mechanism (*Fernandes et al., 2017*). In this view, however, shorter mRNAs would be preferentially impaired by a reduction in 40S recycling, whereas we observed that mRNAs down-regulated in the *tmaΔΔ* mutant tend to be longer than average. In fact, if the predicted deleterious effect of reduced 40S recycling from stop codon to start codon on short mRNAs acts to dampen the stimulatory effect of limiting 43S PICs on their translation, then this could help explain why the *tmaΔΔ* mutations confer an overall weaker reprogramming of TEs compared to SM treatment, given that eIF2α phosphorylation reduces 43S PICs without affecting recycling.

We also considered the possibility that TE changes conferred specifically by elimination of the Tma factors could arise from altered reinitiation following uORF translation, as mammalian MCT-1/ DENR were shown to stimulate reinitiation in a reconstituted system, and depletion of these factors in animal cells reduces reinitiation on certain mRNAs. In yeast, by contrast, elimination of these proteins in combination with eIF2D (Tma64) had the opposite effect of enhancing reinitiation following termination at uORFs of reporter mRNAs in cell extracts (*Young et al., 2018*). However, based on our analysis of *GCN4* translational control, and from quantifying the ratios of ribosome densities in uORFs versus downstream CDSs (RRO values), we uncovered no evidence that eliminating the Tma proteins significantly perturbs reinitiation following uORF translation in yeast cells. Additional experiments using other indicators of reinitiation should be undertaken to examine this issue further. Recently, it was shown that elimination of DENR (Tma22) and eIF2D (Tma64) impairs translation of *ATF4* mRNA both in human cells (*Bohlen et al., 2020*) and in *Drosophila* (*Vasudevan et al., 2020*), which is regulated by the delayed reinitiation mechanism established for yeast *GCN4* mRNA. It appears that loss of DENR/eIF2D reduces reinitiation following translation of *ATF4* uORFs 1-2 owing to a failure to dissociate tRNAs from 40S post-TCs at the penultimate codons of these uORFs (*Bohlen et al., 2020*). Presumably, the penultimate codons in *GCN4* uORFs 1-2 do not require this activity of the yeast recycling factors.

There were many mRNAs whose TEs were affected specifically by depletion of Rps26 and 40S subunit levels. One possible contribution to these specific changes might be the elevated ratio of free 60S to free 40S subunits present only in this condition. This could have the effect of increasing rates of 60S subunit joining to 48S PICs at start codons by mass action, and thereby increase the relative TEs of a subset of mRNAs in which subunit joining is a rate-limiting step of initiation. Some support for this idea comes from our findings here, and from previous results (*Mittal et al., 2017*), that depleting 40S subunits can impair *GCN4* translation, which might result from enhanced subunit joining at the inhibitory uORFs that, in turn, reduces the amount of reinitiation downstream at the *GCN4* CDS. Another possibility is that the aberrant-free ribosomal subunit ratios activate a signaling pathway that alters the translation of particular mRNAs in a manner distinct from the influence of limiting 43S PICs. Clearly, more work is required to achieve a full understanding of the numerous condition-specific TE changes.

Our finding that depletion of Rps26 increases the relative TEs of the group of highly-translated SCL mRNAs was also observed in profiling data from Gilbert et al. (*Thompson et al., 2016*) for two mutants lacking a different ribosomal protein belonging to the large or small subunit, in accordance with the Lodish model. However, findings ostensibly at odds with the model were presented recently (*Ferretti et al., 2017*) involving two groups of yeast mRNAs found to bind preferentially to 40S subunits either containing Rps26 (868 "+Rps26" mRNAs) or lacking Rps26 (743 "ΔRps26" mRNAs). The "+Rps26" mRNAs tend to be more highly translated in WT cells versus the "ΔRps26" mRNAs, and thus are enriched in 'strong' mRNAs; however, five representative mRNAs of the "+Rps26" group showed a greater reduction in translation on Rps26 depletion in comparison to eight representative mRNAs of the "ΔRps26" group. This last finding departs from the prediction of the Lodish model that strong mRNAs should be translated proportionally better than weak mRNAs on reduction of 40S subunit levels by depletion of Rps26. However, by interrogating our own $\Delta TE_{rps26\Delta\Delta}$ values for the "+Rps26" and "ΔRps26" groups of mRNAs, we found an increase in median TE for the "+Rps26" group, and a decrease in median TE for the "ΔRps26" group, (*Figure 5—figure supplement 4*), in keeping with the Lodish model. It is unclear whether the different conclusions regarding the effects of Rps26 depletion on translation of the "+Rps26" and "ΔRps26" mRNAs in our two studies reflect differences in the strains or culture conditions (undefined rich medium there versus defined medium here), the use of polysome association (there) versus ribosome density (here) to assess TE changes, or the particular subsets of representative "+Rps26" and "ΔRps26" mRNAs interrogated for TE changes by *Ferretti et al., 2017*.

A recent study by *Khajuria et al., 2018* on the consequences of reducing the levels of ribosomal subunits in human hematopoietic stem and progenitor cells (HSPCs), regarded as a model of Diamond-Blackfan anemia (DBA), also revealed a widespread reprogramming of translation quite different than observed here in the yeast *rps26ΔΔ* mutant. The mRNAs showing TE reductions in that study tended to be highly translated in untreated HSPCs, and to exhibit short CDSs and short 5'UTRs lacking uORFs, which seems at odds with the predictions of the Lodish model. Although we cannot account for this discrepancy, we note that the global TE values in the untreated HSPCs did

not show the marked inverse correlation with CDS lengths that has been observed previously in yeast (including our study) (*Arava et al., 2003*; *Thompson et al., 2016*), and highly conserved throughout Eukarya (*Fernandes et al., 2017*), including mouse and HeLa cells (*Thompson et al., 2016*). We also note that examining the profiling data of Khajuria et al. indicates that the TEs of RPG mRNAs in untreated HSPCs are well below average, but were found to be above average in a study of HeLa cells (*Guo et al., 2010*; *Guo et al., 2015*), just as they are in yeast. The lower than average TEs of RPG mRNAs in HSPC cells places the reductions in their TEs conferred by reduced ribosome levels in line with the Lodish model; however, the fact that the entire cohort of translationally down-regulated mRNAs have a greater than average median TE in untreated HSPCs seems to suggest that other changes in translation factors or regulatory proteins occur in response to ribosome deple-tion in HSPCs that override the predicted broad outcomes of the Lodish model. As noted above, other factors also appear to modulate the response to limiting 43S PICs in our *rps26ΔΔ* mutant to produce extensive condition-specific TE changes.

At odds with our results on the *rps26ΔΔ* mutant, a study examining the effects of deleting many different RPGs in yeast reached the conclusion that genes for which translation was consistently lower in ribosome-deficient yeast cells had moderately shorter than average CDS lengths, whereas genes for which translation was consistently higher had dramatically longer CDS lengths (*Cheng et al., 2019*). However, using the Ribo-seq and RNA-seq data from this study to calculate TE changes, we observed no clear relationship between TE changes and CDS lengths conferred by the two 40S RPG deletions with the strongest reductions in free 40S levels (*rps29bΔ* and *rps0bΔ*) compa-rable in degree to the depletion of 40S subunits observed in our *rps26ΔΔ* mutant. Compared to the two WT strains analyzed in that study, the *rps29bΔ* and *rps0bΔ* mutants do not exhibit reduced rela-tive TEs for the 20% of all mRNAs with the shortest CDS lengths (*Figure 5—figure supplement 5A*, 1st pentile); nor for the group of RPG mRNAs, which have very short CDS lengths and high basal TE values in WT cells (*Figure 5—figure supplement 5B*, RPG mRNAs). Whereas *rps0bΔ* confers a small increase in relative TE for the pentile of mRNAs with longest CDS lengths (*Figure 5—figure supple-ment 5A*, 5th pentile), this mutation conferred a comparable increase for all mRNAs (*Figure 5—fig-ure supplement 5B*, all mRNAs). Thus, although these results are incongruent with our findings on the *rps26ΔΔ* mutant, they also differ from the findings of *Khajuria et al., 2018* on the effects of depleting ribosomal subunits in human HSPCs.

In summary, our findings suggest that reducing 43S PICs in yeast either by eliminating the 40S ribosome recycling factors, inducing eIF2α phosphorylation, or depleting a 40S subunit protein, all contribute to a similar realignment of TEs wherein inherently strong mRNAs tend to undergo an increase in relative TE at the expense of weak mRNAs. As these trends are predicted by the Lodish model, which should apply generally to all eukaryotic cells, they might be expected to occur in the context of human mutations that reduce the concentration of ribosomal subunits (ribosomopathies) or that reduce the abundance or function of general initiation factors. As pointed out previously (*Mills and Green, 2017*), it may be possible therefore to produce broad changes in the TEs of many cellular mRNAs without the presence of specific regulatory sequences in the sensitive mRNAs by lowering the abundance of the initiation machinery and intensifying the competition between strong and weak mRNAs. It should be kept in mind, however, that other regulatory mechanisms present in mammalian cells but lacking in yeast could alter the outcome from the expectations of the Lodish model for TE changes in ribosomopathies, such as mTORC-mediated control of RPG mRNA transla-tion via 5'-terminal oligopyrimidine (5'TOP) sequences (*Meyuhas and Kahan, 2015*). It is also impor-tant to realize that an increase in relative TE does not necessarily result in higher translational output, as it may coincide with a reduction in absolute TE in a situation where the median translation rate is dramatically lowered; or if it is buffered by a reduction in mRNA abundance, as observed here for the RPG mRNAs in amino acid starved yeast cells. If the function of a critical gene product is highly dose-sensitive, then a small decrease in absolute translational output could be pathological even if the relative TE of its mRNA is elevated in a disease state that involves attenuated 43S PIC assembly.

## Materials and methods

### Key resources table

| Reagent type (species) or resource | Designation | Source or reference | Identifiers | Additional information |
|---|---|---|---|---|
| Strain, strain background (*Saccharomyces cerevisiae*) | MATa his3-Δ1 leu2-Δ0 met15Δ0 ura3Δ0 | Research Genetics | BY4741 | |
| Strain, strain background (*Saccharomyces cerevisiae*) | MATa gcn4::kanMX4 his3Δ1 leu2Δ0 met15Δ0 ura3Δ0 | Research Genetics | 249 | |
| Strain, strain background (*Saccharomyces cerevisiae*) | MATa his3Δ1 leu2Δ0 met15Δ0 ura3Δ0 tma64Δ::kanMX4 | Research Genetics | 4051 | |
| Strain, strain background (*Saccharomyces cerevisiae*) | MATa his3Δ1 leu2Δ0 met15Δ0 ura3Δ0 tma22Δ::kanMX4 | Research Genetics | 6812 | |
| Strain, strain background (*Saccharomyces cerevisiae*) | MATa his3Δ1 leu2Δ0 met15Δ0 ura3Δ0 tma20Δ::kanMX4 | Research Genetics | 328 | |
| Strain, strain background (*Saccharomyces cerevisiae*) | MATa his3Δ1 leu2Δ0 met15Δ0 ura3Δ0 tma64Δ::hygMX4 tma20Δ::kanMX4 | *Young et al., 2018* | H4520 | |
| Strain, strain background (*Saccharomyces cerevisiae*) | MATa his3Δ1 leu2Δ0 met15Δ0 ura3Δ0 tma64Δ::hygMX4 tma20Δ::kanMX5 | *Young et al., 2018* | H4521 | |
| Strain, strain background (*Saccharomyces cerevisiae*) | MATa ura3-52 trp1-Δ63 leu2-3,−112 his4-303 | *Valásek et al., 2004* | H2994 | |
| Strain, strain background (*Saccharomyces cerevisiae*) | MATa ura3-52 leu2-3 leu2-112 trp1-Δ63 his4-303(AUU) kanMX6::$_P$GAL1-RPS26A rps26bΔ::hphMX6 | This study | JVY09 | |
| Recombinant DNA reagent | FLUC reporter with 70nt synthetic 5'UTR with (CAA)n repeats in YCplac33 | *Sen et al., 2016* | FJZ526 | Plasmid |
| Recombinant DNA reagent | FLUC reporter with cap-proximal SL with ΔG of −10.5 kcal/mol in synthetic 5'UTR in YCplac33 | *Sen et al., 2016* | FJZ683 | Plasmid |

*Continued on next page*

*Continued*

| Reagent type (species) or resource | Designation | Source or reference | Identifiers | Additional information |
|---|---|---|---|---|
| Recombinant DNA reagent | FLUC reporter with cap-proximal SL with ΔG of −5.7 kcal/mol in synthetic 5'UTR in YCplac33 | *Sen et al., 2016* | FJZ685 | Plasmid |
| Recombinant DNA reagent | FLUC reporter with cap-distal SL with ΔG of −10.5 kcal/mol in synthetic 5'UTR in YCplac33 | *Sen et al., 2016* | FJZ688 | Plasmid |
| Recombinant DNA reagent | FLUC reporter with cap-distal SL with ΔG of −5.7 kcal/mol in synthetic 5'UTR in YCplac33 | *Sen et al., 2016* | FJZ690 | Plasmid |
| Recombinant DNA reagent | *GCN4-lacZ* (uORFs1-4) in YCp50 | *Hinnebusch, 1985* | p180 | Plasmid |
| Chemical compound, drug | Cycloheximide | Sigma | Cat# C7698 | |
| Chemical compound, drug | Complete EDTA-free Protease Inhibitor cocktail Tablet | Roche | Cat# 1873580001 | |
| Chemical compound, drug | Sulfometuron Methyl | CHEM SERVICE | Cat# PS-1074 | |
| Other | RNase I | Ambion | Cat# AM2294 | enzyme |
| Other | T4 RNA ligase 2 | New England Biolabs | Cat# M0242L | enzyme |
| Other | T4 Polynucleotide kinase | New England Biolabs | Cat# M0201L | enzyme |
| Other | Superscript III | Invitrogen | Cat# 18080044 | enzyme |
| Other | Circ Ligase ssDNA Ligase | Epicentre | Cat# CL4111K | enzyme |
| Other | T4 Rnl2(tr) K227Q | New England Biolabs | Cat# M0351S | enzyme |
| Other | RecJ exonuclease | Epicentre | Cat# RJ411250 | enzyme |
| Other | Yeast 5'-deadenylase | New England Biolabs | Cat# M0331S | enzyme |
| Other | Protoscript II | New England Biolabs | Cat# M0368L | enzyme |
| Other | CircLigase II | Epicentre | Cat# CL9025K | enzyme |
| Other | Phusion polymerase (F-530) | New England Biolabs | Cat# M0530S | enzyme |
| Other | GlycoBlue | Invitrogen | Cat# AM9516 | Blue dye covalently linked to glycogen |
| Commercial assay, kit | fragmentation reagent | Ambion | Cat# AM8740 | |

*Continued on next page*

*Continued*

| Reagent type (species) or resource | Designation | Source or reference | Identifiers | Additional information |
|---|---|---|---|---|
| Commercial assay, kit | miRNeasy Mini Kit | Qiagen | Cat# 217004 | |
| Commercial assay, kit | Ribo-Zero Gold rRNA Removal Kit | Illumina | Cat# MRZ11124C | |
| Commercial assay, kit | Bioanalyzer using the High Sensitivity DNA Kit | Agilent | Cat# 5067–4626 | |
| Commercial assay, kit | Oligo Clean and Concentrator column | Zymo Research | Cat# D4060 | |
| Other | Notched box-plots | http://shiny.chemgrid.org/boxplotr/ | | Web-based tool |
| Other | Gene ontology analysis | https://biit.cs.ut.ee/gprofiler/gost | | Web-based tool |
| Other | Spearman's correlation calculator | https://www.wessa.net/rwasp | | Web-based tool |
| Software, algorithm | ggplot2, R package | https://cran.r-project.org/web/packages/ggplot2/index.html | | |
| Software, algorithm | GeneOverlap, R package | https://www.bioconductor.org/packages/release/bioc/html/GeneOverlap.html | | |
| Software, algorithm | Bowtie | *Langmead et al., 2009* | | |
| Software, algorithm | TopHat | *Trapnell et al., 2009* | | |
| Software, algorithm | Hisat2 | https://github.com/DaehwanKimLab/hisat2 | | |
| Software, algorithm | R script for DESeq2, R package | https://github.com/hzhanghenry/RiboProR | | |
| Software, algorithm | Integrative Genomics Viewer | *Robinson et al., 2011* | | |
| Software, algorithm | Weblogo analysis | *Crooks et al., 2004* | | |

## Strains and plasmids

Wild-type yeast strain BY4741 (*MATa his3-Δ1 leu2Δ0 met15Δ0 ura3Δ0*) and mutant strains 249 (*MATa gcn4::kanMX4 his3Δ1 leu2Δ0 met15Δ0 ura3Δ0*), 4051 (*MATa his3Δ1 leu2Δ0 met15Δ0 ura3Δ0 tma64Δ::kanMX4*), 6812 (*MATa his3Δ1 leu2Δ0 met15Δ0 ura3Δ0 tma22Δ::kanMX4*), and 328 (*MATa his3Δ1 leu2Δ0 met15Δ0 ura3Δ0 tma20Δ::kanMX4*) were purchased from Research Genetics. H4520 (*tma64Δ/tma20Δ*) and H4521 (*tma64Δ/tma22Δ*) were constructed previously (*Young et al., 2018*). Wild-type strain H2994 (*MATa ura3-52 trp1Δ63 leu2-3,−112 his4-303*) (*Valásek et al., 2004*) was the parental strain for JVY09 (*MATa ura3-52 leu2-3 leu2-112 trp1-Δ63 his4-303(AUU) kanMX6::P$_{GAL1}$-RPS26A rps26bΔ::hphMX6*), which was constructed as follows. Strain JVY05 was generated from H2994 by the one-step PCR strategy (*Longtine et al., 1998*), using the *hphMX6* cassette to replace

*RPS26B* and selecting for resistance to hygromycin on rich medium containing galactose as a carbon source (YPGal). JVY09 was similarly generated from JVY05 by the one-step PCR strategy to insert the $P_{GAL1}$ promoter immediately upstream of the *RPS26A* ORF using the *kanMX6* cassette and selecting for resistance to kanamycin on YPGal. Replacement of *RPS26B* with the *hphMX6* cassette and integration of the *kanMX::P_{GAL1}* promoter cassette at *RPS26A* were verified by PCR analyses of genomic DNA using the appropriate primers. JVY09 was shown to be inviable on glucose medium (where the *GAL1* promoter is repressed) in a manner fully complemented by plasmid-borne *RPS26A* on pJVB06.

High-copy *LEU2* plasmid pJVB06 was constructed by inserting into pRS315 (*Sikorski and Hieter, 1989*) a 2.3 kb BamHI restriction fragment containing *RPS26A* flanked by 1120 bp upstream and 832 bp downstream of the coding sequences, amplified from genomic DNA of strain H2994, creating pJVB05. The insert from pJVB05 was then subcloned into pRS425 (*Sikorski and Hieter, 1989*) to generate pJVB06. Plasmid p180 containing the *GCN4-lacZ* reporter was described (*Hinne-busch, 1985*), as were luciferase reporter plasmids FJZ526, FJZ683, FJZ685, FJZ688, and FJZ690 (*Sen et al., 2016*).

## Polysome profile analysis

For strains BY4741 (WT), H4520 (*tma64Δ/tma20Δ*), and 4051 (*tma64Δ*), log-phase cultures of $A_{600}$ = 0.5–0.6 were harvested by centrifugation and whole cell extracts (WCEs) were prepared by vortexing the cell pellet with two volumes of glass beads in ice cold 1X breaking buffer (20 mM Tris-HCl (pH 7.5), 50 mM KCl, 10 mM MgCl$_2$, 1 mM DTT, 200 μg/mL heparin, 50 μg/mL cycloheximide (CHX), and 1 Complete EDTA-free Protease Inhibitor cocktail Tablet (Roche)/50 mL buffer). Twenty $A_{260}$ units of WCEs were layered on a pre-chilled 10–50% sucrose gradient prepared in 1X breaking buffer using a BioComp Gradient Master (BioComp Instruments), according to the manufacturer's instructions, and centrifuged at 40,000 rpm for 2 hr at 4°C in a SW41Ti rotor (Beckman). Continuous $A_{260}$ scanning of the fractionated gradient was conducted using the BioComp Gradient Station (Bio-comp Instruments, Canada). Analysis of strains H2994 (WT) and JVY09 (*rps26ΔΔ*) was conducted identically except that log-phase cultures were harvested at $A_{600}$ = 0.4–0.5, and WCEs were layered on a pre-chilled 5–45% sucrose gradient.

## Preparation of ribosome footprint libraries

Ribosome profiling and RNA-seq analysis were conducted in parallel, primarily as described previously (*Zeidan et al., 2018*), using isogenic strains BY4741 (WT), H4520 (*tma64Δ/tma20Δ*), and 4051 (*tma64Δ*); or H2994 (WT) and JVY09 (*rps26ΔΔ*), or 249 (*gcn4Δ*) with two biological replicates performed for each genotype. Strains BY4741 and H4520 were grown at 30°C in synthetic complete medium (SC) to log-phase ($A_{600}$ = 0.5–0.6). In parallel, BY4741 was grown in synthetic complete media lacking isoleucine and valine (SC-Ile/Val) to log-phase ($A_{600}$ = 0.5–0.6) and treated with sulfo-meturon methyl (SM) at 1 μg/mL for 25 min to induce eIF2α phosphorylation. Cells were fast-filtered and snap-frozen in liquid nitrogen, and lysed in a freezer mill in the presence of lysis buffer (20 mM Tris (pH 8), 140 mM KCl, 1.5 mM MgCl$_2$, 1%Triton X-100, 500 μg/mL CHX). For ribosome-protected mRNA fragment (RPF) library preparation, 60 $A_{260}$ units of cell lysates were digested with RNase I (Ambion; AM2294) at 15U per $A_{260}$ unit for 1 hr at room temperature (25°C) on a Thermomixer at 700 rpm; and then resolved by sedimentation through a 10–50% sucrose gradient to isolate 80S monosomes. RPFs were purified from monosomes using hot phenol-chloroform, and 25–34 nt fragments were size-selected after electrophoresis through a 15% TBE-Urea gel. The purified RPFs were ligated to universal miRNA cloning linker (Synthesized by Integrated DNA Technologies) using trun-cated T4 RNA ligase 2 (New England Biolabs; M0242L) after a dephosphorylation reaction carried out with T4 Polynucleotide kinase (PNK; New England Biolabs, M0201L). The ligated RPF products were size-selected after electrophoresis on a 10% TBE-Urea gel, reverse-transcribed using Super-script III (Invitrogen; 18080044), size-selected again, and circularized using CircLigase ssDNA Ligase (Epicentre; CL4111K). Ribosomal RNA contamination was reduced by oligonucleotide subtraction hybridization. Each 'subtracted' library was amplified by PCR to add unique six nt indexes and com-mon Illumina primer and flow cell binding regions. Quality of the libraries was assessed with a Bioa-nalyzer using the High Sensitivity DNA Kit (Agilent 5067–4626) and quantified by Qubit. Sequencing

was done on an Illumina HiSeq system at the NHLBI DNA Sequencing and Genomics Core at NIH (Bethesda, MD).

For RNA-seq library preparation, total RNA was extracted and purified from aliquots of the same snapped-frozen cells described above using hot phenol-chloroform extraction. Five μg of total RNA was randomly fragmented by mixing with 2× alkaline fragmentation solution (2 mM EDTA, 10 mM $Na_2CO_3$, 90 mM $NaHCO_3$, pH ≈ 9.3). After incubation for 20 min at 95℃, fragmentation reactions were mixed with ice-cold stop solution (300 mM NaOAc pH 5.5, 30 μg GlycoBlue [Invitrogen; AM9516]). RNA was precipitated by adding one part of isopropanol followed by the standard proto-col of precipitation. Fragment size selection, library generation, and sequencing were carried out using the same protocol described above for RPF library preparation, except the Ribo-Zero Gold rRNA Removal Kit (Illumina; MRZ11124C), was employed to remove rRNA after linker-ligation.

H2994 and JVY09 were cultured at 30℃ to $A_{600}$ of 0.8 O.D. in SC medium containing 2% galac-tose and 2% raffinose in place of glucose. Cells were pelleted and shifted to glucose-containing SC medium for 3 hr before harvesting at $A_{600}$ = 0.4–0.5. RPF and RNA sequencing libraries were con-structed as described above except that, for RPF library preparation, 35 $A_{260}$ units of cell lysates were subjected to RNase I digestion, and for RNA sequencing libraries, total RNA extraction was carried out using a miRNeasy Mini Kit (Qiagen; 217004) from 35 $A_{260}$ units of the same extracts used for RPF preparation.

Strains BY4741 and 4051 were grown at 30℃ in SC to log-phase ($A_{600}$ = 0.5–0.6). The cells were snap-frozen and 35 $A_{260}$ units of lysates were used to construct RPF libraries, as described above. For RNA sequencing library preparation a modified protocol was used, as previously described (*McGlincy and Ingolia, 2017*). In short, after RNA extraction, five μg of total RNA was randomly fragmented using fragmentation reagent (Ambion; AM8740). After incubation for 12 min at 70℃, fragmentation reactions were mixed with stop solution (Ambion; AM8740). Fragment size selection was carried out using the same protocol described above for RPF library preparation. Size selected RNA fragments were ligated to preadenylated sample-specific barcode linker (Synthesized by Inte-grated DNA Technologies) using T4 Rnl2(tr) K227Q (New England Biolabs; M0351S) after a dephos-phorylation reaction carried out by T4 PNK (New England Biolabs; M0201S). The unligated linker was specifically depleted from the ligation reaction by Yeast 5'-deadenylase and RecJ exonuclease 10 U/ul (Epicentre; RJ411250). The samples were pooled and purified using Oligo Clean and Con-centrator column (Zymo Research; D4060). Further, the Ribo-Zero Gold rRNA Removal Kit (Illumina; MRZ11124C), was employed to remove rRNAs. The rRNA depleted samples were reverse-tran-scribed using Protoscript II (New England Biolabs; M0368L) and subsequently circularized by CircLi-gase II (Epicentre; CL9025K). Each 'circularized' library was amplified by using NI-798 and NI-799, forward and reverse primers (Integrated DNA Technologies), respectively, using Phusion polymerase (F-530) (New England Biolabs; M0530S), and gel-purified. Quality and quantity of the libraries were assessed as described above. Sequencing was done on an Illumina HiSeq system at the NHLBI DNA Sequencing and Genomics Core at NIH (Bethesda, MD).

Strain 249 was grown at 30℃ in SC media to log-phase ($A_{600}$ = 0.5–0.6), and in parallel cultured in SC-Ile/Val to log-phase ($A_{600}$ = 0.5–0.6) and treated with SM at 1 μg/mL for 25 min to induce eIF2α phosphorylation. RPF sequencing libraries were constructed with 35 $A_{260}$ units of cell lysates as described above for RNA sequencing library preparation using the modified protocol except that for RPF library preparation, rRNA contamination was reduced by oligonucleotide subtraction hybrid-ization. For RNA sequencing libraries, total RNA extraction was carried out using a miRNeasy Mini Kit (Qiagen; 217004) and library construction was carried by the NHLBI core sequencing facility using the TruSeq Stranded mRNA Library Prep Kit (Illumina).

## Alignment of RPF and RNA reads and visualization in a genome browser

As described earlier (*Martin-Marcos et al., 2017*), Illumina sequencing reads were trimmed, and the trimmed FASTA sequences were aligned to the *S. cerevisiae* ribosomal database using Bowtie (*Langmead et al., 2009*). The non-rRNA reads (unaligned reads) were then mapped to the *S. cerevi-siae* genome using TopHat (*Trapnell et al., 2009*). Only uniquely mapped reads from the final geno-mic alignment were used for subsequent analyses. Wiggle files were generated as described previously (*Zeidan et al., 2018*) and visualized using Integrative Genomics Viewer (IGV 2.4.14, http://software.broadinstitute.org/software/igv/) (*Robinson et al., 2011*). Wiggle tracks shown are

normalized according to the total number of mapped reads. To visualize the 80S ribosomes stalled ~30 nt upstream of stop codon, only uniquely mapped reads were aligned to stop codons as described previously (*Ingolia et al., 2014*). The normalized reads were used to generate plots in Microsoft excel. Analysis of RNA sequencing libraries for strains BY4741 and 4051, and RPF sequencing libraries for strain 249, was conducted identically except that PCR duplicates were removed from trimmed sequencing reads and these sequences were aligned to the *S. cerevisiae* ribosomal database using Bowtie to remove ncRNA (*Langmead et al., 2009*). RNA sequencing libraries of strain 249 were mapped to the genome using Hisat2 available at GitHub https://github.com/DaehwanKim-Lab/hisat2; *Gaikwad, 2021*; copy archived at swh:1:rev:1ec47602507e7237ab85603255b7cfa954763a66.

## Differential gene expression analysis of ribosome profiling data

Statistical analysis of changes in mRNA, RPFs, or TE values between two biological replicates from two genotypes or culture conditions being compared was conducted using DESeq2 (*Love et al., 2014*), excluding any genes with less than 10 total mRNA reads in the four samples (of two replicates each) combined. DESeq2 is well-suited to identifying changes in mRNA or RPF expression, or TEs, with very low incidence of false positives using results from only two highly correlated biological replicates for each of the strains/conditions being compared (*Zhang et al., 2014*; *Lamarre et al., 2018*). Briefly, DESeq2 pools information from the thousands of genes analyzed to model count variances for genes of similar expression levels, which are used in a generalized linear model (GLM) to identify expression changes and place confidence intervals on the magnitude of changes, and exclude genes with aberrantly high variability. Transcriptional and translational changes are analyzed together in a GLM by including library type (mRNA-seq or Ribo-seq) as one of the variables, in addition to genotype/condition, in a multi-factor design. TE emerges as the effect of Ribo-seq library type against the mRNA-seq baseline, and significant interactions of TE with the genotype/condition indicate translational control (*Ingolia, 2016*). The R script employed for DESeq2 analysis of TE changes can be found on Github (https://github.com/hzhanghenry/RiboProR; *Kim et al., 2019*).

## Analysis of uORF translation using ribosome profiling data

Relative ribosome occupancies (RRO) of translated uORFs were calculated as described previously (*Martin-Marcos et al., 2017*). Briefly, translated uORFs in annotated 5′-UTRs were identified using the yassour-uorf program (*Brar et al., 2012*), wherein the ratio of RPF counts at the +1 position (uORF start codon) to −1 position was calculated for all putative uORFs initiating with either AUG or near-cognate codons. All uORFs with +1 /−1 RPF ratios > 4, combined RPF counts at the +1 and −1 positions >14, and >50% of the RPF counts in the zero frame with respect to the start codon were selected for further analysis. These selection criteria were applied to multiple ribosome profiling datasets to obtain a set of 6061 translated uORFs of at least three codons in length, including 564 AUG-initiated and 5497 near-cognate-initiated uORFs. The uORF identification tool RibORF (*Ji et al., 2015*) was applied as a second filter, which is based on three nucleotide periodicity and uniformity of read distribution across uORF codons, using a probability threshold of 0.5, to obtain a final list of 2721 uORFs with evidence of translation in the profiling datasets from which they were first identified using the yassour-uorf program. A bed file containing the sequence coordinates of the 2721 uORFs was combined with bed files containing the coordinates of the 5′UTR, main CDS, and 3′UTR of each gene, and used to obtain RPF counts for the uORFs and associated main CDS in each strain examined, excluding the first and last nucleotide triplets of the 5′UTRs, the first and last codons of the uORFs, and the first 20 codons of the CDS. The RRO for each uORF was calculated by dividing RPF counts in the uORF by RPF counts of the main CDS; and statistical analysis of changes in RRO values between the two biological replicates of two different genotypes was conducted using DESeq2 (*Love et al., 2014*), excluding uORFs with mean RPF counts <2 or mean CDS RPF counts <32 in the four samples (of two replicates each) combined. A similar analysis of RRO values was performed on a set of evolutionarily conserved uORFs identified by McManus et al. (*Spealman et al., 2018*).

## Statistical and gene ontology analyses

All notched box-plots were constructed using a web-based tool at http://shiny.chemgrid.org/box-plotr/ and their significance was accessed by the Mann-Whitney U test computed using the R Stats package in R. Spearman's correlation analysis was conducted using the online tool at https://www.wessa.net/rwasp_spearman.wasp. Gene ontology (GO) categories determined for each set of genes were identified using 'g:Profiler' (https://biit.cs.ut.ee/gprofiler/gost). Enrichment of nucleotides upstream and downstream of the start codon was determined by Weblogo analysis (*Crooks et al., 2004*). Hierarchical cluster analysis of TE changes in mutants was conducted with the R heatmap.2 function from the R 'gplots' library, using the default hclust hierarchical clustering algorithm. The Kolmogorov-Smirnov test, used to evaluate the cumulative distributions in cumulative distribution and frequency distribution plots, was computed using the R Stats package in R. Significance of gene set overlaps in Venn diagrams was evaluated with the hypergeometric distribution by Fisher's exact test computed in R using the GeneOverlap package. Pearson's correlation and student's t-tests analysis were done using the formula imbedded in Microsoft Excel. Scatterplots displaying correlations between read counts from replicates of ribosome profiling or RNA-seq experiments, and volcano plots, were created using the scatterplot function in Microsoft Excel. Cumulative distribution and frequency distribution plots were generated using the frequency function in Microsoft excel. Smoothed scatterplots used to examine the correlation between RRO changes in the *tma*ΔΔ mutant versus WT cells were computed and plotted using the ggplot2 package in R.

## Data resources for mRNA attributes

Yeast 5'UTR and CDS lengths were taken from *Pelechano et al., 2013*, WT mRNA steady-state levels (in molecules per dry cellular weight - pgDW) were obtained from *Lahtvee et al., 2017*, and steady-state mRNA half-lives were obtained from *Neymotin et al., 2014*. The AUG context score (context score) (*Miyasaka, 1999*) was calculated as $A_{UG}CAI = (w_{-3} \times w_{-2} \times w_{-1} \times w_{+4})^{1/4}$ where $w_i$ is the fractional occurrence of that particular base, normalized to the most prevalent base, present in the $i^{th}$ position of the context among the ~270 most highly expressed yeast genes, taken from the matrix of frequencies and relative adaptiveness (w) of the nucleotide in the AUG context of this group of ~270 reference genes (*Zur and Tuller, 2013*).

## Assaying *GCN4-lacZ* expression

Specific activities of β-galactosidase activity in WCEs were determined as previously described (*Moehle and Hinnebusch, 1991*). Enzyme activities are expressed in units of nmoles of o-nitro-phenyl-β-D-galactopyranoside (ONPG) cleaved per min per mg of total protein.

## Luciferase assays

Overnight-grown cultures of transformants of strains BY4741 (WT) and H4520 (*tma64Δ/tma20Δ*) harboring the appropriate reporter plasmids were diluted and grown for approximately three cell doublings in SC-Ura at 30°C. Three mL of exponentially growing cells were lysed with glass beads in 400 µL of ice-cold lysis buffer (1× PBS containing one Complete EDTA-free Protease Inhibitor Cocktail Tablet (Roche)/50 mL). WCEs were cleared by centrifugation and luciferase activity (relative light units, RLUs) was measured in 5 µL of 1:10 diluted WCEs using the Promega Luciferase Assay System and a luminometer (Berthold Technologies). Undiluted WCEs were used for total protein concentration measurements using the Bradford Reagent (BioRad) and known amounts of bovine serum albumin as standards. RLUs were normalized by the total protein amounts.

## Acknowledgements

We thank Fujun Zhou for providing us with uORF lists, scatterplot scripts, and yeast CDS length, 5'UTR length and context score databases. We acknowledge Shardul Kulkarni, Fan Zhang, and Sara Young-Baird for assistance in data analysis. We thank Thomas Dever, Nicholas Guydosh and Jon Lorsch for many helpful discussions about data analysis and interpretation of results, and all members of the Hinnebusch, Lorsch, Dever, and Guydosh labs, for useful comments.

## Additional information

### Competing interests

Jyothsna Visweswaraiah: Jyosthna Visweswaraiah is affiliated with Pandion Therapeutics; however, the author has no financial interests to declare. The other authors declare that no competing interests exist.

### Funding

| Funder | Grant reference number | Author |
|---|---|---|
| Intramural Research Program of the NIH | HD001004-37 | Alan G Hinnebusch |

The funders had no role in study design, data collection and interpretation, or the decision to submit the work for publication.

### Author contributions

Swati Gaikwad, Conceptualization, Formal analysis, Investigation, Writing - original draft, Writing - review and editing; Fardin Ghobakhlou, Formal analysis, Investigation, Writing - review and editing; David J Young, Jyothsna Visweswaraiah, Investigation, Writing - review and editing; Hongen Zhang, Software, Formal analysis; Alan G Hinnebusch, Conceptualization, Formal analysis, Supervision, Funding acquisition, Project administration, Writing - review and editing

### Author ORCIDs

Swati Gaikwad  https://orcid.org/0000-0002-1438-9497
Hongen Zhang  http://orcid.org/0000-0001-6871-8463
Alan G Hinnebusch  https://orcid.org/0000-0002-1627-8395

### Decision letter and Author response

Decision letter https://doi.org/10.7554/eLife.64283.sa1
Author response https://doi.org/10.7554/eLife.64283.sa2

# Additional files

### Supplementary files

• Transparent reporting form

### Data availability

Sequencing data from this study have been deposited to the NCBI Gene Expression Omnibus (GEO) under GEO accession number GSE166987.

The following dataset was generated:

| Author(s) | Year | Dataset title | Dataset URL | Database and Identifier |
|---|---|---|---|---|
| Gaikwad S, Ghobakhlou F, Young DJ, Visweswaraiah J, Zhang H, Hinnebusch AG | 2020 | Reprogramming of translation in yeast cells impaired for ribosome recycling favors short, efficiently translated mRNAs | https://www.ncbi.nlm.nih.gov/geo/query/acc.cgi?acc=GSE166987 | NCBI Gene Expression Omnibus, GSE166987 |

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
