## [Decision Letter]

[Editors' note: this paper was reviewed by Review Commons.]

**Acceptance summary:**

The high-resolution biochemical experiments reported here demonstrate that defects in recycling ribosomes that have finished synthesizing a protein result in the efficient translation of a subset of messenger RNAs that have been previously shown to be efficiently translated under conditions in which initiation of translation is impaired. The results have implications for our understanding of the translational control of gene expression and, possibly, our understanding of human diseases caused by defects in the concentration of ribosomes available for initiation.

---

## [Author Response]

Reviewer #1 (Evidence, reproducibility and clarity (Required)):Summary:The manuscript by Gaikwad et al. describes a study into the role of yeast Tma64 and Tma20/22 in post-termination ribosome recycling. The central hypothesis is that as yeast strains deleted for the respective genes are known to show defects in ribosome recycling, such strains might also show a reduction in functional ribosomal subunits. This hypothesis is tested by monitoring the translational efficiency of the yeast transcriptome using Ribo-Seq and RNA-Seq approaches, together with polysome profiles and reporter assays. The central conclusion is that they do indeed observe specific effects on the translational machinery which they attribute to impaired post-termination ribosome recycling, as opposed to the ablation of putative translation initiation factor activities of Tma64 and Tma20/22.Major comments:Although the central hypothesis is clearly formulated and makes sense, the relationship between the experimental results and the hypothesis is unclear and a number of results are implicitly or explicitly presented as supporting the hypothesis, although it is not clear to me why. Specifically:1) While the polysome profiles support the existence of some translational defect in the deletion strains, I do not think that they are consistent with a reduction in 43S PIC levels. First, there is no evidence of a change in free 40S/60S ratios which I would expect to see if 40S recycling was impaired to the extent that this would result in functional consequences.

This is a good point. We have calculated the free 40S/60S ratios and indeed found no significant decrease in the *tmaΔΔ* mutant. In addition, it is expected that a fraction of these unrecycled 40S subunits will have moved from the stop codon and are scanning the 3’UTR to generate the increased 3’UTR translation previously demonstrated in *tmaΔΔ* cells (Young et al., 2018) and confirmed here (Figure 1B). It is likely that such 40S post-termination complexes located either at stop codons or scanning in 3’UTRs will dissociate from mRNA during polysome fractionation by sucrose density gradient centrifugation because they lack base-pairing between a bound tRNA and an mRNA codon—a property which necessitated the use of formaldehyde fixation to study scanning 40S subunits by the TCP-Seq method for small-subunit profiling (Archer, S. K., et al. (2016). Nature 535(7613): 570-574), and this dissociation would prevent us from observing the predicted reduction in free 40S subunits. We have also proposed that sequestration of 40S subunits in the 80S ribosomes engaged in abnormal 3’UTR translation in *tmaΔΔ* cells, which occurs on essentially all 3’UTRs (Young et al., 2018), also contributes to the reduced availability of free 40S subunits for PIC assembly in this mutant; and this sequestration also will not reduce the free 40S/60S ratio. It is also relevant that the *tma20Δ* deletion was shown to confer an elevated free 40S/60S ratio, which may result from a 60S biogenesis defect (Fleischer et al., 2006); if so, this could offset the depletion of free 40S subunits expected from accumulation of unrecycled 40S subunits at stop codons and in 3’UTRs. All of these points have been added to the Discussion to provide a plausible explanation for the absence of a diminished ratio of free 40S to 60S subunits in *tmaΔΔ* cells. Moreover, we have added a statement in the Discussion that we cannot eliminate the possibility that the impaired ribosome recycling in *tmaΔΔ* cells is not the major defect responsible for the reprogramming of translation with the features characteristic of a reduction in 43S PIC assembly. We admit that the complete loss of eIF2-independent Met-tRNAi recruitment could be involved, or that a defect in ribosome biogenesis may result in aberrant 40S subunits in addition to reducing 60S subunit levels, either of which could be exacerbated by the defective 40S recycling in this strain.

Second, the authors observe "halfmer" peaks indicative of small ribosomal subunits that are stuck at some point between transcript binding by the small subunit and large subunit joining. The halfmer intermediate is after the 43S PIC stage when the small subunit has bound the transcript, so in my view is not evidence for reduced 43S PIC levels. Consistent with this view, the RP deletion strain which serves as a reference for actual reduced 43S PIC levels does not show these half-mer peaks.

While it is true that halfmers result from a defect in 60S subunit joining to late-stage 48S PICs containing initiator tRNA base-paired to AUG start codons, they could also result, as we had suggested, from accumulation of unrecycled 40S subunits at stop codons, as both defects will yield an extra 40S subunit bound to mRNAs containing translating 80S ribosomes. However, since we expect the unrecycled 40S subunits at stop codons or scanning in 3’UTRs to dissociate from the mRNA during sedimentation (as just described above), we agree that the halfmers we observe in the tmaΔΔ mutant more likely arise from the 60S biogenesis defect conferred by tma20Δ with attendant accumulation of 48S PICs at AUG start codons. Accordingly, we have removed the claim that halfmers provide evidence corroborating the recycling defect in the tmaΔΔ mutant and rely instead on the queuing of 80S ribosomes upstream of stop codons and the elevated translating 80S ribosomes in 3’UTRs (both shown in Figure 1A-B) as evidence confirming the known recycling defect in the tmaΔΔ mutant.

2) The detailed analyses of translational efficiency in tmaΔΔ strains are interesting, but the connection to post-termination recycling is established via comparisons to control strains which I do not fully follow. A comparison of the tmaΔΔ strain with a tma64 single deletion is used to make the case that the double deletion asserts its effects through a recycling defect rather than a translation initiation defect. This test is based on the observation that the mammalian TMA64 homologue can act as a translation factor. An implicit assumption in this experiment is that Tma64 and Tma20/22 have homologous activities in recycling, but only Tma64 could potentially act in translation initiation. What is the rationale for this assumption?

This is a good point, as the MCT-1/DENR heterodimer was also shown to function in eIF2-independent Met-tRNAi recruitment to AUG codons when the AUG was placed in the P site by a viral IRES (Skabkin et al., 2010). Realizing this, we were careful to limit our conclusion by saying that eliminating a function for eIF2D in initiation does not lead to re-programming of translation in the manner observed in the *tmaΔΔ* strain. While we think this evidence is valuable to include in the paper, we have qualified its interpretation by stating that it does not rule out the possibility that eIF2D or MCT-1/DENR are fully redundant for the putative eIF2-independent Met-tRNAi recruitment in vivo such that MCT-1/DENR would be sufficient for a completely WT level of this activity in cells lacking eIF2D.

3) For the metagene profiles in Figure 1A it is stated in the text that the WT profile shows "the expected three nucleotide periodicity and a peak at the stop codon". In the tmaΔΔ strains, the stop codon peak is ablated but a new peak appears -30 to -40 nt upstream of the stop codon. The transition from the stop codon peak to the upstream peak is one of the main pieces of evidence for the recycling defect of the tmaΔΔ strain. In the experiment in Figure 3B which displays the tma64 control, the wild-type strain and the tma64 deletion control have neither a stop-codon peak nor a -30 peak, which makes me wonder whether this experiment was actually done under conditions comparable to the tmaΔΔ profiling.

The absence of a peak at -30 nt upstream of the stop codon is expected, as neither the WT strain nor *tma64Δ* mutant have recycling defects. In our experience, the presence of a pronounced 80S peak at stop codons varies for wild-type strains from one profiling experiment to the next, perhaps reflecting the lability of unrecycled 80S post-termination complexes in vitro. As such, the presence of translating 80S ribosomes queued upstream from stop codons and the elevated 3’UTR occupancies of translating 80S ribosomes—both stabilized by cycloheximide addition to the extracts—are the key indicators of a defect in 40S recycling at stop codons, which we have observed reproducibly in the *tmaΔΔ* strain (Figure 1A-B), but not in the *tma64Δ* single mutant (Figure 3B-C). We have modified the Results to include this explanation.

4) The tmaΔΔ strain is compared to an amino acid starvation condition ("+SM") known to lead to eIF2 phosphorylation that reduces the activity of this translation initiation factor, and later to a ribosomal protein deletion which serves as a reference for 40S subunit depletion. The TE changes in the tmaΔΔ strain appear to resemble the translation initiation defect more closely than 40S subunit depletion (Figure 5D), yet the authors conclude that the main effect of the tmaΔΔ strain is a recycling defect. This does not make sense to me.

As pointed out in the text, all three conditions exhibit a similar reprogramming of TEs in which inherently strong mRNAs tend to show increased TEs at the expense of weak mRNAs. We have also clearly noted the appearance of TE changes that are limited to only one of the three conditions. Although these condition-specific differences between the *tmaΔΔ* mutant and the other two conditions are relatively greater for the *rps26ΔΔ* versus SM-treated WT cells, this does not invalidate the conclusion that the *tmaΔΔ* mutant shows TE changes indicative of increased competition for limiting 43S PICs that favors inherently stronger mRNAs. We have discussed possible reasons for condition-specific TE changes in the Discussion, and acknowledged that more work is required to achieve a comprehensive accounting of them.

In sum of the major points, I remain unconvinced that the TE changes observed in the study really stem from recycling defects.

Given the strong evidence published previously by Young et al., 2018, there is little doubt that 40S recycling is impaired in the *tmaΔΔ* mutant. We have found no evidence for altered reinitiation following translation of uORFs in this mutant, suggesting that the other function for MCT-1/DENR established by in vivo studies in mammalian cells in regulating reinitiation following translation of certain uORFs, is not a significant driver of TE changes in the *tmaΔΔ* mutant. The only other viable possibility is that these proteins provide an alternative to eIF2 for recruitment of Met-tRNA to ribosomes. As noted above, having found no TE changes nor recycling defects in the *tma64Δ* mutant lacking eIF2D, combined with the ability of eIF2D to function on its own in Met-tRNAi recruitment in vitro established by others, the simplest interpretation of the data is that the recycling defect observed in *tmaΔΔ* cells contributes to the TE changes observed in the double mutant. Nevertheless, as noted above, we have added a sentence in the Discussion indicating that a 40S recycling defect and attendant sequestration of 40S subunits at stop codons and in scanning or translating 3’UTRs may not be sufficient to account for the translational reprogramming and that some other defect in the *tmaΔΔ* strain also contributes, which might include a ribosome biogenesis defect conferred by *tma20Δ.*

Minor comments:5) What is the difference between Figure 1A and supplemental figure 2? If the supplemental figure shows an exact experimental replication of the main text figure, I'm not sure this add anything to the manuscript.

Presumably, the reviewer meant Figure 1C versus 1A; in which case, our response would be that we think it is helpful to the reader to include in the supplemental figure the results of gradient separations for biological replicates to bolster the results presented in Figure 1C; although as noted above, we will revise our interpretation of the halfmers and focus only on the reductions in polysome:monosome ratios in the *tmaΔΔ* strain.

6) Why the switch from cumulative fraction format to box plots in Figure 2D? I think it would be easier for a reader if all sub-panels were in the same format (cumulative fractions work fine).

These are equally valid ways to demonstrate statistically significant differences in the median values for two sets of data.

Reviewer #1 (Significance (Required)):If all the conclusions of the paper were sound, this study would corroborate the function of Tma64 and Tma20/22 in ribosome recycling, and would demonstrate that efficient ribosome recycling via these factors is required to maintain a sufficient supply of fresh 43S PICs to maintain functional initiation rates. As it stands, the study describes extensive ribosome profiling work with a number of conditions that impair aspects of protein synthesis, but with the connection to ribosome recycling poorly supported in my view I am not sure what the actual scientific message would be.

It was well established by Young et al., 2018, that there is a 40S recycling defect in the *tmaΔΔ* mutant, and while corroborated here, this is not a major conclusion of our work. Rather, our analysis of TE changes in this mutant, not analyzed previously, indicate a distinctive signature in which mRNAs enriched for all of the attributes of well-translated mRNAs, including greater than average TEs in WT cells, tend to show TE increases in the mutant. As this trend is predicted by theory for reductions in the availability of 43S PICs, and was also observed in the two other conditions of eIF2α phosphorylation and depletion of bulk 40S subunits analyzed here, we think it is justified to propose that a reduction in 43S PICs is an important (but not necessarily the sole) driver of the reprogramming of TEs that accompanies defective 40S recycling in the *tmaΔΔ* mutant. In choosing the title of the paper, we have been careful to emphasize the results that were experimentally determined, and to indicate in the Abstract that the results are consistent with a reduction in 43S PICs being an important driver of the major trend identified in the TE reprogramming observed in the *tmaΔΔ* mutant. Nevertheless, we have altered the title of the paper to read “Reprogramming of translation in yeast cells impaired for ribosome recycling favors short, efficiently translated mRNAs”.

Reviewers Cross-CommentingI note that my review was considerably less favourable than that of the other two reviewers. Any comments on the validity of particular points I made would be welcome.I support the request made by reviewer 3 to make analysis scripts available.Reviewer 3: The argument regarding alternative hypotheses seems sound. I think this adds to an overall picture where the conclusions are not quite consistent with the data.Reviewer #2 (Evidence, reproducibility and clarity (Required)):In this work, Gaikwad et al. investigate the effects of impaired small subunit recycling and reduced initiation rates on the translatome of *Saccharomyces cerevisiae*. Previous work by the authors showed that depletion of the Tma proteins Tma64, Tma20, and Tma22 led to defects in small subunit recycling and reinitiation in 3'UTRs. Building off that study, the authors asked if depletion of free 40S subunits led to significant reprogramming in translated genes due to reduced initiation, namely, reduced levels of available 43S PICs. Using ribosome profiling, the authors found a significant subset of genes that were affected by loss of Tma64 and Tma20. To ascertain if these effects were due to changes in initiation, the authors also conducted ribosome profiling of cells that were starvation induced using sulfometuron methyl (SM). They found a significant overlap between affected genes in both conditions, indicating that reduced initiation was primarily responsible for these effects, although several genes showed marked differences. To modulate initiation using a different approach, the authors also conducted ribosome profiling in a rps26 depletion mutant. Again, the results showed overlaps between the three conditions with a significant number of genes also showing marked differences. From these data the authors concluded that translational reprogramming was skewed to favor "strong" transcripts – those transcripts that are abundant and efficiently translated – at the cost of "weak" transcripts, and that the responsible mechanism was changes in competition for 43S PICs due to a reduction in free 40S subunits.The data appear of high quality and support the main conclusions of the paper.

We were gratified to see that this reviewer found our results to be both of high quality and supportive of our main conclusions, and had only minor concerns.

I have a few minor concerns however:1) Among the three different conditions tested, there were unique genes whose changes in TE did not match with others. The authors noted that the differences in the mechanisms of reduced ternary complex (TC) levels, impaired ribosome recycling, or altered ratios of free 60S to 40S subunits could act differently on specific mRNAs. This is likely to be true; for example, reduced TC clearly affects translation of genes with uORFs like GCN4, while the other two deficiencies do not. Arguably, however, this can also complicate analysis since some of these genes can have global effects on transcription (the denominator when measuring TEs). Taking GCN4 as an example, its induction leads to dramatic changes in transcript levels and in particular affects the very same genes whose TEs are used to make the main conclusion of the paper. It would be nice if the authors conducted ribosome profiling in the presence of SM, but in a GCN4-deletion strain to offset this effect.

It should be noted that all three conditions we analyzed lead to a down-regulation of mRNA levels and up-regulation of TEs for the ribosomal protein gene (RPG) mRNAs, a major group of highly translated, strong closed-loop mRNAs. As such, an increase in TE that buffers the decrease in abundance for these mRNAs was found in all three conditions and is not unique to the SM-treatment that induces Gcn4. Nevertheless, we have conducted ribosome profiling of SM-treated isogenic *gcn4Δ* cells and found that the characteristic reprogramming of TEs observed in SM-treated WT cells remains evident in the absence of the transcriptional changes induced by Gcn4. These new data have been incorporated into a new supplementary figure (Figure 2—figure supplement 2D) and described in Results.

2) Another additional analysis that would strengthen the authors conclusions might be to analyze ribosomal profiling data of yeast subjected to other stressors (eg. Oxidative stress, glucose starvation, heat shock, etc.). Some of these are known to also inhibit initiation through various mechanisms. While those studies will not be done under the same conditions as this work, analyzing responsive genes in various contexts could further inform the ways in which these genes maintain translation even under conditions where initiation is altered. It could also highlight trends in the attributes of these transcripts or the pathways in which they operate.

We have analyzed other published ribosome profiling datasets to determine whether or not we observe a similar reprogramming of translation as observed in the three conditions we analyzed, involving rapamycin and diamide treatment of WT cells isogenic to the strains analyzed here. Although both conditions are reported to induce eIF2α phosphorylation, we observed induction of *GCN4* translation only in the rapamycin data set. These two conditions might also impact initiation in other ways, including down-regulation of eIF4G abundance or eIF4F assembly (at least for rapamycin), which could alter the pattern of translational reprogramming for mRNAs for which some other aspect of initiation is rate-limiting besides recruitment of 43S PICs to activated mRNAs. The results shown in Author response image 1 indicate that neither of these two stress conditions closely resembles the pattern of translational reprogramming we observed in all three of our conditions, with the *tmaΔΔ* mutant data shown for comparison. Because we don’t know all of the steps of initiation that might be impacted by these two conditions, it’s difficult to draw any firm conclusions from the results, and we have elected not to include these data in the revised paper.

**Author response image 1. sa2fig1:** 

Reviewer #2 (Significance (Required)):Overall, the manuscript provides significant insights into how changes to translation initiation through multiple mechanisms can drastically alter translation efficiencies and cause translational reprogramming. These could have important ramifications for understanding how certain ribosomopathies can have seemingly specific effects on gene expression.Reviewer #3 (Evidence, reproducibility and clarity (Required)):SummaryThis manuscript examines the effect of impaired ribosome recycling on the translation of endogenous mRNAs in the budding yeast, *Saccharomyces cerevisiae*. They use the genome-wide techniques of ribosome profiling (Ribo-seq) and RNA-seq to estimate the relative translation efficiency of each mRNA (TE: Ribo-seq counts / RNA-seq counts) and compare this quantity between mutant strains and wild-type strain. Their first major finding is that a large group of mRNAs with high baseline TE in the wild-type strain also have positive changes in TE (ΔTE) in the mutant strain with impaired ribosome recycling. These mRNAs with high baseline TE also have short CDS and 5'UTR in comparison with other mRNAs, and have been previously implicated (Costello, 2015) in strong closed loop formation through association with eIF4E, eIF4G and PABP. Their second major finding is that two other perturbations, decrease in 43S complex formation through eIF2α phosphorylation and depletion of the Rps26 ribosomal protein, also exhibit the same trend as the ribosome recycling mutant with regards to the high baseline TE mRNAs having positive ΔTE. They conclude that the high baseline TE directly drives the positive ΔTE when the availability of ribosomes for translation initiation is limited by different perturbations. This conclusion is based on a previous kinetic model and measurements by Lodish, 1974 showing that protein synthesis from well-translated β-globin mRNA is more robust against ribosome concentration decrease than protein synthesis from the less-translated α-globin mRNA. The authors suggest that their observations might be applicable to human ribosomopathy disorders which also exhibit a decrease in cellular ribosome concentration.Major comments1) The authors' key conclusion is that under conditions of depleted 43S complex concentration, mRNAs with inherently high rates of translation efficiency are favored for translation over other mRNAs. I am not convinced by this conclusion since it based on just correlation between ΔTE and TE without any other experiment that controls for the many confounding variables. For example, as the authors themselves note (Figure S4), ΔTE is just as strongly correlated with CDS length as it is with basal TE in all three of their key experiments (first vs. last bar in each panel in Figure S4). The authors do not present any experimental evidence that favors high TE over CDS length. It will be more accurate to present an unbiased discussion of the different possible driver parameters instead of favoring TE over other parameters.

We agree that it is formally possible that CDS length is a more fundamental driver of TE changes than the basal TE in the WT control cells, but this would beg the question of the molecular mechanism involved. Shorter mRNAs might be more capable of forming the closed-loop intermediate, which could enhance eIF4F association owing to eIF4G-PABP interaction, or more capable of recycling of ribosomes *in cis* from the stop codon to start codon; and both features would help them to compete for limiting 43S PIC at the initiation step. It is also important to note that all of features known to be associated with high basal TE in WT cells are also associated with increased TE in our three mutants/conditions, including elevated propensity for closed-loop formation, higher than average Kozak context scores, higher than average mRNA abundance, and longer than average mRNA half-lives. (The non-intuitive correlation between TE and mRNA abundance likely reflects the fact that efficiently expressed genes are both highly transcribed and endowed with multiple mRNA features that optimize them for translation, including a short CDS.) Given the highly conserved inverse relationship between CDS length and TE across many different species, it is likely that our finding that short CDS is a predictor of enhanced TE in all three mutant/stress conditions we examined reflects the fact that mRNAs with short CDSs tend to be inherently stronger mRNAs better able to compete for limiting PICs during initiation. We have now treated this issue thoroughly in the Discussion.

To substantiate the claim that TE and not some other variable such as CDS length is driving the observed ΔTE, the authors should perform reporter experiments where one of these variables is changed while the other is held constant (for example, see Thompson, 2016 Figure 3E who conclude that CDS length is critical for ΔTE when the RP Asc1 is deleted). The authors can vary TE while holding length constant and measure ΔTE in different mutant backgrounds. These experiments should not take more than a few weeks to conduct given the expertise of the authors. In fact, they use similar reporter experiments to study uORF-mediated control in this manuscript (Figure 7A).

For all of the reasons mentioned above regarding the strong, conserved inverse correlation between CDS length and basal TE, it is likely impossible to vary the CDS length while holding TE constant in a set of reporter constructs. We also note that for the CDS length reporters analyzed by Thompson et al. cited by the reviewer, the protocol for quantifying their translation does not allow measurement of the individual TEs for long and short CDS reporters in each strain, only the ratios of TE in mutant/WT for each reporter. As such, it cannot be ascertained whether the reporter with longer CDS actually shows a lower basal TE in WT compared to the shorter CDS reporter. In addition, these reporters require induction by galactose, which is incompatible with the culture conditions we employed in our profiling experiments. The additional suggestion for reporter assays (as we understand it) to analyze *GCN4-lacZ* reporters with or without uORFs as a way of varying TE while holding CDS length fixed seems to overlook the fact that this regulation by uORFs is known to be exquisitely sensitive to levels of the eIF2-GTP-Met-tRNAi ternary complex, which governs the rate of reassembly of scanning 43S PICs following termination at uORF1. We have seen in our ribosome profiling data that SM induction in WT induces *GCN4* translation (as expected), the *tmaΔΔ* mutation has no effect, and the *rps26ΔΔ* mutation reduces *GCN4* translation, which likely results from enhanced 60S subunit joining at the inhibitory uORFs owing to the elevated free 60S concentration (as we discussed). Thus, the three conditions we analyzed can have differential effects on translation of an mRNA with particular cis-acting regulatory elements, such as the *GCN4* uORFs, independently of their common effects in reducing 43S PIC concentrations discernible by examining large ensembles of mRNAs by ribosome profiling. As an alternative to the reviewer’s suggestion, however, we have analyzed the effect of the *tmaΔΔ* mutations on expression of a panel of luciferase reporters containing or lacking insertions of stem-loop structures of differing stabilities in the 5’UTR situated either close to the cap or at a cap-distal position. Importantly, we found that the *tmaΔΔ* mutation substantially reduced expression only of the reporters harboring a cap-proximal stem-loop, predicted to impede PIC attachment. These new data, presented in a new supplementary figure (Figure 6—figure supplement 2), support the idea that the *tmaΔΔ* mutation preferentially reduces the translation of poorly translated mRNAs with a relatively greater effect on mRNAs expected to be impaired for 43S PIC loading.

2) The relevance to ribosomopathies is highly speculative and likely not to hold given the well-known features of mammalian translation. In mammals, ribosomal protein mRNAs are down-regulated in comparison to other mRNAs due to their 5'TOP motifs when eIF2α is phosphorylated (see Sidrauski, 2015). This is the opposite of what the authors observe in yeast. The effect of ribosome depletion on translation of mammalian mRNAs is unclear (see contrasting claims by Khajuria, 2017 and Tiu, 2020).

While it is true that Sidrauski et al., 2015, observed a modest down-regulation of the TEs of a subset of RPG mRNAs when eIF2 is phosphorylated, they concluded that this might not involve the 5’TOP motifs of these mRNAs and the inhibition of mTORC since the effect was reversed by the eIF2B activator ISRIB and thus apparently results from eIF2α phosphorylation. Nevertheless, the regulation of RPG mRNAs in mammalian cells might not conform to the situation we observed in yeast owing to other regulatory circuits like TORC-mediated control via TOP sequences. We also agree that the reprogramming of TE changes we observed in *rps26ΔΔ* yeast cells is distinct from that reported for 40S protein depletions studied by Khajuria et al., 2017; which we have already treated in detail in our Discussion. We think it is worth mentioning that our results in yeast are in fact consistent with the ribosome concentration/competition model for TE changes, and that this model should be considered in evaluating TE changes in ribosomopathies, as proposed already by Eric Mills and Rachel Green, 2017, especially since there are conflicting results in different studies of mammalian 40S protein depletions. Nevertheless, we have added a statement to the Discussion indicating that other regulatory mechanisms present in mammalian cells but lacking in yeast, such as TORC-mediated control via TOP sequences, could alter the outcome from the expectations of the ribosome concentration model for TE changes.

3) I highly recommend providing the code used to generate the plots from raw data (or from TE tables) even if it not well organized. Why makes it needlessly hard for an interested reader to reproduce the findings and figures?

We have posted on Github the R code employed for DESeq2 analysis of TE changes, and included the link in the revised paper. All other calculations were conducted using the on-line tools we have cited, or with Microsoft Excel.

4) Experimental replicates and statistical analysis are mostly adequate (except for the minor comment below).Minor comments1) This study does not cite or compare their results to a relevant recent study Cheng, 2019 where ribosome profiling was performed on a large group of RPG deletion mutants.

We have added a new supplementary figure (Figure 5—figure supplement 5) and an accompanying paragraph to the Discussion discussing the results in the Cheng et al., 2019, paper. This study concluded, in agreement with the findings of Khajuria et al., 2018, for human cells, that “genes for which translation was consistently lower in RP-deficient yeast cells were moderately shorter than the overall spectrum of ORF lengths, and that the genes for which translation was consistently higher in RP-deficient cells were dramatically and significantly longer than the overall spectrum of ORF lengths”. However, using the Ribo-Seq and mRNA-Seq data from this study to calculate TE changes, we observed no clear relationship between TE changes and CDS lengths conferred by the two 40S RPG deletions with the strongest reductions in free 40S levels, *rps29bΔ* and *rps0bΔ,* comparable to the depletion of 40S subunits observed in our *rps26ΔΔ* mutant. Compared to the two WT strains analyzed in that study, the *rps29bΔ* and *rps0bΔ* mutants do not exhibit reduced relative TEs for the 20% of all mRNAs with the shortest CDS (Figure 5—figure supplement 5A, 1^st^ pentile); nor for the group of RPG mRNAs, which have very short CDSs and high basal TE values in WT cells (Figure 5—figure supplement 5B, RPG mRNAs). Whereas *rps0bΔ* confers a small increase in relative TE for the pentile of mRNAs with longest CDSs (Figure 5—figure supplement 5A, 5th pentile), this mutation conferred a comparable increase for all mRNAs (Figure 5—figure supplement 5B, all mRNAs). Thus, although these results are incongruent with our findings on the *rps26ΔΔ* mutant, they also differ from the findings of Khajuria et al., 2018, on the effects of depleting ribosomal subunits in human HSPCs. As pointed out already in our Results section, our conclusion that limiting 43S PICs in the three conditions we studied upregulates the TEs of the high-TE group of strong closed loop mRNAs, which have shorter than average CDS lengths and are highly enriched for the RPGs, is also supported by data from Thompson et al., 2016 who reported upregulation of the TEs of SCL mRNAs in two mutants lacking different ribosomal proteins. We’re not sure why the results in Cheng et al. on the RPG mRNAs do not agree more strongly with ours.

2) The authors have many sentences and even a section title that refer to increase in translation efficiency or translation upregulation in their experiments. Translation efficiency of most mRNAs (perhaps except GCN4) likely does not actually increase under any of these conditions. The authors note this point in their discussion, but it will be more accurate and helpful for a non-expert reader if it is discussed early in the Results section or the authors conclusions are re-worded to be more accurate. I recommend calling the Ribo-seq / RNA-seq ratio as RTE (relative translation efficiency between genes) instead of TE to avoid misinterpretation.

We agree and have added text at the first mention of TE to indicate that relative TE values are being reported throughout; and also added the term “relative” to section headings and at relevant places in the main text that contain the word “TE”.

3) The authors use Venn Diagrams to show that changes in mRNA abundance and TE are not correlated (Figure 1E, 4C). These Venn Diagrams do not allow the reader to evaluate the raw data nor do they have any statistical significance. It will be more transparent and robust if the authors plot the fold-changes in mRNA and TE against each other as a scatter plot (at least for all the TE-changing mRNAs if not all mRNAs) and indicate the correlation coefficient and associated P-value.

We have added the P values to the Venn diagrams calculated using Fisher's exact test and have shown the values for significant over-enrichment compared to expectation by chance. This was the case only for one of the four comparisons in both figures, supporting our conclusion that the TE changes are generally not associated with mRNA changes.

Reviewer #3 (Significance (Required)):The authors finding of coherent translational changes of closed-loop mRNAs under seemingly very different mechanistic perturbations (impaired ribosome recycling, eIF2α phosphorylation, ribosomal protein depletion) is an important and novel finding. This finding in itself will be of broad interest to the translation community. However, extensive discussion of the ribosome concentration hypothesis when it is just one possible interpretation of their data tempered my enthusiasm for this otherwise well-conceived and executed project. Rather than strongly implying that the observed results might be relevant to ribosomopathies, a more nuanced discussion of potential differences between the yeast and mammalian experiments models (while also including a discussion of Cheng, 2019) will serve to enlighten rather than confuse the non-expert reader.

We were gratified that this reviewer also found our evidence convincing in revealing coherent translational re-programming in response to three disparate conditions that should negatively impact 43S PIC assembly, and also judged the findings to be important, novel and of broad interest to the translation field. As indicated above, we have added a discussion of the alternative hypothesis that CDS length per se is a driver of TE re-programming and explain why we favor the idea that it is the highly conserved, inverse relationship between CDS length and inherent TE that is responsible for the tendency of short mRNAs to be upregulated under conditions of limiting 43S PICs, which can be rationalized by the ribosome concentration hypothesis. We have also tempered our claims that our findings are relevant to an understanding of TE changes in ribosomopathies, and note only that our findings in yeast are generally consistent with the ribosome concentration hypothesis, and cite published literature in which this hypothesis has been considered in examining the mechanism of translation changes in these human diseases. We have also noted that there are significant differences between the mechanism of translation initiation and cis-acting regulatory sequences in mRNAs between yeast and mammals, that might distinguish our findings from those in mammalian cells.

As noted above, we have included in the Discussion a comparison of our findings and those reported by Cheng et al., 2019, on genome-wide changes in translation conferred by deletions of different ribosomal proteins. Analyzing their data for the two 40S protein deletions with the strongest effects on cell growth and depletion of free 40S subunits, and thus most similar to our *rps26ΔΔ* strain, we found that their results do not appear to support their stated conclusion that, “in agreement with the findings of Khajuria et al., 2018, genes for which translation was consistently lower in RP-deficient yeast cells were moderately shorter than the overall spectrum of ORF lengths, and that the genes for which translation was consistently higher in RP-deficient cells were dramatically and significantly longer than the overall spectrum of ORF lengths (Figure 3F).” Thus, while the Chang et al. findings do not show the tendency for short, high-TE mRNAs like those of the RPGs to exhibit increased TE on depletion of 40S subunits, it is not the case that they show the exact opposite trends described in mammalian cells by Khajuria et al., 2018,. We don’t understand the differences between our two sets of findings but feel it is important that our data and analysis be published for comparison to the previously published findings.

Reviewers Cross-CommentingI agree with reviewer #1's comment about the unchanged 40S/60S ratio in the tmaΔΔ strain leading to the question of whether 43S availability is really reduced in this strain.What do the other 2 reviewers think about the central claim in the Title and Abstract that efficiently translated mRNAs are translationally favored in the different mutant strains? Their data is completely consistent with the alternate conclusion that "short mRNAs are translationally favored in the mutant strains". The authors favor the former over the latter but I do not see any evidence supporting this.

In deference to this reviewer, we have changed the title to: “Reprogramming of mRNA translation in yeast cells impaired for ribosome recycling favors short, efficiently translated mRNAs.”